# LoRAPrune: Pruning Meets Low-Rank Parameter-Efficient Fine-Tuning

## Abstract

Large pre-trained models (LPMs), such as LLaMA and GLM, have shown exceptional performance across various tasks through fine-tuning. Although low-rank adaption (LoRA) has emerged to cheaply fine-tune these LPMs on downstream tasks, their deployment is still hindered by the vast model scale and computational costs. Neural network pruning offers a way to compress LPMs. However, the current pruning methods designed for LPMs are not compatible with LoRA. This is due to their utilization of unstructured pruning on LPMs, impeding the merging of LoRA weights, or their dependence on the gradients of pre-trained weights to guide pruning, which can impose significant memory overhead. To this end, we propose LoRAPrune, a new framework that delivers an accurate, compact model for efficient inference in a highly memory-effective manner. Specifically, we first design a LoRA-guided pruning criterion, which uses the weights and gradients of LoRA, rather than the gradients of pre-trained weights for importance estimation. We then propose a structured iterative pruning procedure, to remove redundant channels and heads. Extensive experimental results demonstrate the superior performance of our LoRAPrune over existing approaches on the LLaMA series models. For instance, at a 50% compression rate, LoRAPrune outperforms LLM-Pruner by a perplexity reduction of 4.81 on WikiText2 and 3.46 on PTB datasets, while concurrently reducing memory usage by 52.6%.

## 1 Introduction

Large pre-trained models (LPMs) (Touvron et al., 2023; Du et al., 2022; Frantar et al., 2023) have showcased remarkable prowess, exhibiting outstanding performance across numerous tasks. To enable LPMs to perform specific tasks, such as chat-bots (Du et al., 2022; Zeng et al., 2022), they are often efficiently fine-tuned on downstream datasets (Taori et al., 2023; Chenghao Fan & Tian, 2023) by parameter-efficient fine-tuning (PEFT) methods (Jia et al., 2022; Hu et al., 2022; Chen et al., 2022), among which LoRA-based fine-tuning methods (Hu et al., 2022; Luo et al., 2023; He et al., 2023) have gained widespread use. However, the remarkable success of LPMs is accompanied by obstacles from their vast scale and substantial computational costs, making deployment exceedingly arduous (Frantar & Alistarh, 2023).

Neural network pruning (Li et al., 2017; Molchanov et al., 2017), a popular technique for model compression, can significantly reduce the model size and complexity. Recently, the post-training pruning literature, such as SparseGPT (Frantar & Alistarh, 2023) and WANDA (Sun et al., 2023), have achieved high-performance unstructured sparse LPMs. However, unstructured sparse models face two critical issues: *1) Unstructured sparse models are hard to obtain direct inference speedup*. They often require specialized hardware support to achieve satisfying acceleration benefits, which leads to unstructured pruning not benefiting legacy off-the-shelf platforms, *e.g.*, CPUs, DSPs, and GPUs (Fang et al., 2023; You et al., 2023; Zhou et al., 2022). *2) Unstructured sparse models are not compatible with LoRA*. As shown in Figure 1 (a), since the weights **BA** produced by LoRA are dense, it poses challenges when trying to merge **BA** into the unstructured sparse

Table 1: The Memory costs for pruning LLaMA-65B. "Iter." indicates whether the method supports iterative pruning and "#GPU" indicates the number of NVIDIA A100 (80G) GPUs required.

| Method | Iter. | #GPU | Mem.(G) |
|---|---|---|---|
| PST (Li et al., 2022b) | ✓ | 3 | 234 |
| LLM-Pruner (Ma et al., 2023) | ✗ | 2 | 154 |
| LoRAPrune | ✓ | 1 | 72 |

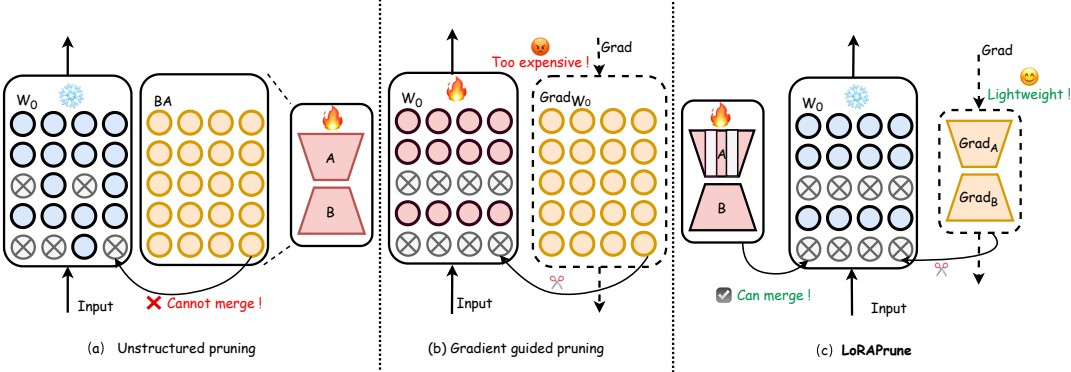

Figure 1: Comparing LoRAPrune with other pruning methods: (a) Unstructured sparse model cannot directly merge LoRA weights, which is computationally inefficient. (b) Gradient-guided pruning requires the gradients of the pre-trained weights, which is memory-intensive. (c) LoRAPrune only needs the gradients of LoRA weights and can seamlessly merge LoRA weights into pre-trained weights, which is efficient in both memory and computation.

weights. LoRA without merging increases inference time by nearly 54% (see Table 3), diminishing the benefits of pruning. One potential solution is to perform fine-tuning using LoRA on downstream tasks first and then carry out post-training pruning. However, separating tuning and pruning can lead to sub-optimal results (Molchanov et al., 2019; Sanh et al., 2020). To tackle this challenge, PST (Li et al., 2022b) combines unstructured pruning with efficient fine-tuning, which simultaneously prunes LoRA and pre-trained weights. This method ensures a seamless merge of LoRA weights and avoids additional computational overhead that comes from LoRA. However, unstructured pruning of LoRA necessitates computing $\mathbf{BA}$ first and then doing Hadamard product with a binary mask $\mathbf{M}$, which results in significant memory overhead (see Table 1) since $\mathbf{BA}$ and $\mathbf{M}$ share the same shape with pre-trained weights. For instance, when pruning LLaMA-65b, the intermediate variables necessitate the storage capacity of three NVIDIA A100 (80G) GPUs. This poses a significant memory challenge when adapting PST to LPMs. Instead, structured pruning can mitigate this issue since we can directly prune the structured weights of $\mathbf{A}$ in LoRA without storing $\mathbf{BA}$. Therefore, it is significant to combine LoRA with structured pruning to achieve simultaneous parameter-efficient fine-tuning and direct acceleration on general hardware platforms with high performance.

To this end, we propose a unified framework for LoRA and structured pruning, named LoRAPrune. As shown in Figure 1 (c), LoRAPrune not only prunes the structured weights (*e.g.*, heads, channels) from the pre-trained model weights $\mathbf{W}_0$ but also trims the corresponding weights in LoRA weight $\mathbf{A}$ without computing $\mathbf{BA}$ first. Consequently, after pruning and fine-tuning, the weights of LoRA can be *seamlessly* merged with the pre-trained weights, ensuring that no additional computations are needed during inference. To identify weight connections of structural importance, the criterion used in the structured pruning methods (Ma et al., 2023; Molchanov et al., 2019; 2017) is often estimated by gradients or its variants, as shown in Figure 1 (b). However, LoRA typically requires frozen pre-trained weights without computing their gradients, thus pruning approaches that rely on gradients of the pre-trained weights cannot be directly applied. To *efficiently* estimate the importance of pre-trained weights, LoRAPrune introduces a new criterion by only employing the gradients of LoRA. In contrast to the vanilla gradient-guided pruning method, LoRAPrune leverages LoRA's gradients as the approximation for the gradients of the pre-trained weight. Based on the presented criterion, we can *iteratively* perform pruning while simultaneously conducting efficient fine-tuning to restore the performance of the pruned LPMs, requiring only a small calibration set. Specifically, we compute the importance of every batch of data and update the importance using a moving average. Every few iterations, we remove a portion of unimportant structured weights until the desired sparsity is achieved. Through extensive experiments on a variety of benchmark datasets and different scale LPMs, we demonstrate that LoRAPrune consistently outperforms other structured pruning techniques tailored for LPMs. Furthermore, compared to the vanilla gradient-guided pruning method, LoRAPrune notably reduces memory and computational consumption, enabling simultaneously efficient pruning and fine-tuning LLaMA-65b on one GPU.

This paper has the following key contributions:

- We introduce a novel weight importance criterion for LPMs, LoRA-guided criterion, that seamlessly works with LoRA. With the gradients of LoRA, we can efficiently approximate the importance of pre-trained weights without needing to compute their gradients.

- Based on the proposed criterion, we introduce LoRAPrune, a new framework that unifies PEFT with pruning. Since we can efficiently approximate the gradients and update weights by LoRA, LoRAPrune allows for iterative structured pruning, which can achieve high compression rates while ensuring that the pruned model can be seamlessly integrated with LoRA weights.

- Pruning experiments conducted on the LLaMA models demonstrate that LoRAPrune can efficiently perform structured pruning with up to 65 billion weights on one GPU. Furthermore, the pruning results achieved by LoRAPrune significantly surpass other pruning methods. For instance, when compared to LLM-Pruner, LoRAPrune demonstrates remarkable efficiency by requiring only 52.6% of the memory overhead while achieving significantly lower perplexity scores on the WikiText2 and PTB datasets, outperforming LLM-Pruner by 4.81 and 3.46 perplexity, respectively.

## 2  RELATED WORK

**Parameter-efficient fine-tuning.** PEFT methods (Jia et al., 2022; Wu et al., 2022; Chen et al., 2022; Hu et al., 2022; Luo et al., 2023; He et al., 2023) have received increasing attention from both academia and industry. Among them, LoRA (Hu et al., 2022) proposes injecting trainable low-rank decomposition matrices into each layer which can be merged into the pre-trained weights, avoiding extra computation in inference. Since inference efficiency, many methods based on LoRA have emerged. For instance, GLoRA (Chavan et al., 2023) facilitates efficient parameter adaptation by employing a scalable, modular, layer-wise structure search that learns individual adapters of each layer. LongLoRA (Chen et al., 2023) improves upon LoRA, enabling efficient fine-tuning of LPMs on long contexts. AnimateDiff (Guo et al., 2023) obtains a personalized generator by inserting LoRA into the frozen text-to-image model. Quantizing the pre-trained weights into 4-bit, QLoRA (Dettmers et al., 2023) employs LoRA for fine-tuning LPMs in downstream tasks while maintaining efficient memory usage. Therefore, LoRA is indispensable for fine-tuning LPMs. Our method seamlessly integrates LoRA and pruning, making it easily extensible to other PEFT methods based on LoRA.

**Neural network pruning.** Removing unimportant weights from LPMs to reduce memory and the computational cost of deployment has become a common approach for model compression. Unstructured pruning (Dong et al., 2017; Lee et al., 2019; Wang et al., 2020; Sun et al., 2023; Frantar & Alistarh, 2023; Li et al., 2022b) can obtain highly compressed models by directly pruning neurons, which also causes unstructured sparsity and hard deployment. In contrast, structured pruning (Molchanov et al., 2019; 2017; Ma et al., 2023; He et al., 2019; Fang et al., 2023) directly discards the whole grouped parameters (*e.g.*heads, channels) and leaves a model with deploy-friendly structures. Our method also focuses on structured pruning, which can directly obtain inference acceleration.

**Pruning criterion.** Determining the importance of weights in a network is still an open question (Blalock et al., 2020). A common approach to model pruning is to use parameter magnitude (Li et al., 2018; Lee et al., 2020; Elesedy et al., 2020; Han et al., 2015; Li et al., 2017) as a criterion. However, the small weights can still have a significant impact on the model output due to the complex structure of neural networks, while large weights may not be as important. Many methods (Molchanov et al., 2017; 2019; Sanh et al., 2020; Yu et al., 2022a; Zhang et al., 2022; Lee et al., 2019; Yu et al., 2022b; Wang et al., 2020; LeCun et al., 1989; Hassibi et al., 1993) employ Taylor expansion to approximate the errors introduced by pruning and use this as the criterion for importance estimation. To avoid computing the Hessian matrix (Hassibi et al., 1993) or Hessian inverse (LeCun et al., 1989) in Taylor expansion, Molchanov et al. (2017; 2019) only use the first-order term in Taylor expansion. Furthermore, LLM-Pruner (Ma et al., 2023) similarly utilizes the first-order expansion for pruning and extends the pruning technique to LPMs. However, the first-order term in Taylor expansion still requires gradients of the pre-trained weights. As shown in Table 1, computing and storing the gradients of pre-trained weights significantly increases the pruning cost. To avoid computing gradients of pre-trained weights, PST (Li et al., 2022b) learns the gradients of pre-trained weights by an extra low-rank matrix, which is motivated by LoRA. Nevertheless, PST conducts unstructured pruning and needs to compute a substantial mask with the equivalent shape of pre-trained weights in each forward pass, which is memory-intensive and hard to be adapted to LPMs. Different from

LLM-Pruner (Ma et al., 2023) and PST (Li et al., 2022b), our criterion only relies on LoRA's gradients and does not require expensive mask computation, making it memory-efficient.

## 3 METHOD

### 3.1 PRELIMINARY

Initially, we define the notation used in the formula. "**Bold**" letters represent matrices and vectors, while "non-bold" letters indicate scalars. "Subscripts" identify the index of elements within a matrix, and "superscripts" indicate the layer index in a network.

**Low-rank adaptation.** To efficiently fine-tune LPMs, low-rank adapter LoRA (Hu et al., 2022) constrains the update of model parameters to maintain a low intrinsic rank. During fine-tuning, the pre-trained weights remain frozen, abstaining from gradient computation, while the inserted LoRA is kept trainable. Given two low-rank matrices $\mathbf{A} \in \mathbb{R}^{r \times k}$ and $\mathbf{B} \in \mathbb{R}^{d \times r}$ ($r \ll \min(d, k)$), the update of a linear module can be written as

$$\mathbf{z} = \mathbf{x}\mathbf{W}_0 + \mathbf{x}\mathbf{B}\mathbf{A}, \tag{1}$$

where $\mathbf{W}_0 \in \mathbb{R}^{d \times k}$, $\mathbf{z} \in \mathbb{R}^{n \times k}$ and $\mathbf{x} \in \mathbb{R}^{n \times d}$ denote the pre-trained weights, outputs and inputs, respectively. After adaption, the new weights $\mathbf{W}$ can be re-parameterized as $\mathbf{W} = \mathbf{W}_0 + \mathbf{B}\mathbf{A}$.

**Pruning with Taylor expansion.** In vanilla pruning approaches (Molchanov et al., 2017; 2019), the importance of a weight $\mathbf{W}_{i,j} \in \mathbf{W}_0$ can be quantified by measuring the impact of its removal on the loss. For an input $\mathbf{x}$ and the ground-truth prediction $\mathbf{y}$, the induced error of $\mathbf{W}_{i,j}$ can be given as:

$$\mathbf{I}_{i,j} = (\mathcal{L}(\mathbf{x}, \mathbf{y}, \mathbf{W}_0) - \mathcal{L}(\mathbf{x}, \mathbf{y}, \mathbf{W}_0 | \mathbf{W}_{i,j} = 0))^2. \tag{2}$$

Computing $\mathbf{I}_{i,j}$ for each weight is computationally expensive. Following Molchanov et al. (2019), we can use first-order Taylor expansion to approximate the importance $\hat{\mathbf{I}}_{i,j}$ by:

$$\hat{\mathbf{I}}_{i,j} = (\frac{\partial \mathcal{L}}{\partial \mathbf{W}_{i,j}} \mathbf{W}_{i,j})^2. \tag{3}$$

**Dependency-aware structured pruning.** In structured pruning, it is crucial to consider that pruned neurons can exhibit dependencies with other neurons due to their interconnected nature. The dependencies of weights are illustrated in Figure 5. We organize the connected weights as a group and estimate the group importance by accumulating the weight importance within the same group. Formally, the importance for the $g$-th group can be expressed as

$$\hat{\mathcal{G}}_g = \sum_{\mathbf{W}_{i,j} \in \mathbb{G}} \hat{\mathbf{I}}_{i,j}, \tag{4}$$

where $\hat{\mathcal{G}} \in \mathbb{R}^{1 \times G}$ represents the importance of groups, $\mathbb{G}$ denotes a set of weights within a group and $G$ is the candidate group number in a layer.

### 3.2 PRUNING WITH LOW-RANK ADAPTION

**Motivation.** To achieve high-compressed LPMs, it is essential to accurately evaluate the importance of pre-trained weights. A key approach is to utilize the criteria in Eq. (3) for this evaluation. However, obtaining the gradient of $\mathbf{W}_0$ in a LPM is difficult since it requires a lot of computing power and storage space. Fine-tuning LPMs with LoRA is becoming prevalent (Taori et al., 2023; Chenghao Fan & Tian, 2023). During LoRA fine-tuning, only the gradients of LoRA's weights are computed, yielding remarkable computation and memory efficiencies. Therefore, can we rely solely on the weights and gradients of LoRA to accurately estimate the importance of pre-trained weights?

**LoRA-guided criterion.** In this work, we discuss how to estimate the importance of $\mathbf{W}_0$ by inserting the learnable matrices $\mathbf{A}$ and $\mathbf{B}$ in the downstream task adaption.

The core idea lies in setting the element $(\mathbf{B}\mathbf{A})_{ij} = -\mathbf{W}_{ij}$ if the element $\mathbf{W}_{ij} \in \mathbf{W}_0$ is removed. The importance of each parameter in Eq. (2) can be reformulated as follows

$$\mathbf{I}_{i,j} = (\mathcal{L}(\mathbf{x}, \mathbf{y}, \mathbf{W}) - \mathcal{L}(\mathbf{x}, \mathbf{y}, \mathbf{W} | (\mathbf{B}\mathbf{A})_{i,j} = -\mathbf{W}_{i,j}))^2. \tag{5}$$

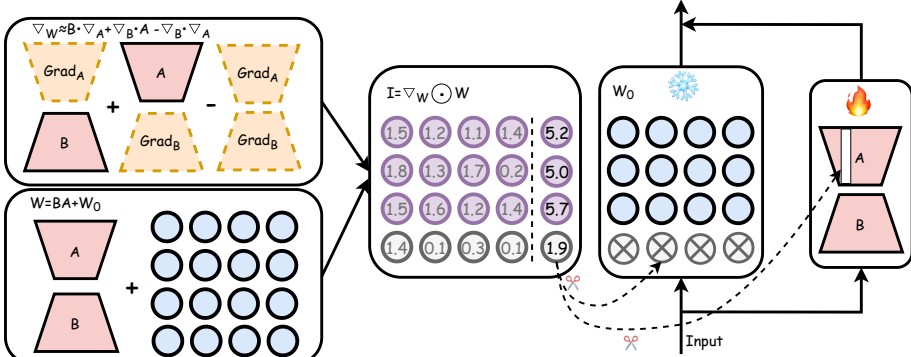

Figure 2: The pruning process for the LoRA-guided criterion involves utilizing the low-rank matrices $\mathbf{A}$, $\mathbf{B}$ and their respective gradients $\nabla_{\mathbf{A}}$, $\nabla_{\mathbf{B}}$ to compute the importance $\mathbf{I}$. Subsequently, weight importance (gray) with the same group are aggregated to the group importance (black) and the groups with low scores are removed.

Exploiting the first-order Taylor expansion with $(\mathbf{BA})_{i,j} = -\mathbf{W}_{i,j}$ to approximate Eq. (5), the estimated importance $\hat{\mathbf{I}}_{i,j}$ of parameter $\mathbf{W}_{i,j}$ can be represented by

$$\hat{\mathbf{I}}_{i,j} = (\frac{\partial \mathcal{L}}{\partial (\mathbf{BA})_{i,j}}((\mathbf{BA})_{i,j} + \mathbf{W}_{i,j}))^2. \tag{6}$$

However, as shown in Eq. (1), the LoRA computation sequence involves first multiplying by $\mathbf{B}$ and then by $\mathbf{A}$, which means that $\mathbf{BA}$ cannot be obtained during the forward and backward pass. Besides, preserving $\frac{\partial \mathcal{L}}{\partial (\mathbf{BA})_{i,j}}$ still entails the same level of complexity as $\frac{\partial \mathcal{L}}{\partial \mathbf{W}_{i,j}}$ since $\mathbf{BA}$ shares the same shape of $\mathbf{W}_0$.

Here, we only save and use the gradients of two low-rank matrices $\mathbf{A}$ and $\mathbf{B}$ to approximate $\frac{\partial \mathcal{L}}{\partial (\mathbf{BA})}$. We can rely on the gradient update that $(\mathbf{BA})_{i,j}|_t = (\mathbf{BA})_{i,j}|_{t-1} - \eta \frac{\partial \mathcal{L}}{\partial (\mathbf{BA})_{i,j}}$ to estimate the gradient, where $(\mathbf{BA})_{i,j}|_t$ and $(\mathbf{BA})_{i,j}|_{t-1}$ represents the $(\mathbf{BA})_{i,j}$ in $t$-th and $(t-1)$-th step, respectively. For simplicity, we ignore the learning rate $\eta$ since it has no influence on the importance criterion. Apparently, $\frac{\partial \mathcal{L}}{\partial (\mathbf{BA})_{i,j}}$ is proportional to the change of $\mathbf{BA}$, which can be written as

$$\frac{\partial \mathcal{L}}{\partial (\mathbf{BA})_{i,j}} \propto [(\mathbf{BA})_{i,j}|_{t-1} - (\mathbf{BA})_{i,j}|_t]. \tag{7}$$

Here, $(\mathbf{BA})_{i,j}|_t = \mathbf{B}_{i,:}|_t \mathbf{A}_{:,j}|_t$ is generated by the multiplication of the $i$-th row of $\mathbf{B}|_t$ with the $j$-th column of $\mathbf{A}|_t$. Using the above assumption, we can also estimate $\frac{\partial \mathcal{L}}{\partial \mathbf{A}_{:,j}} \propto \mathbf{A}_{:,j}|_{t-1} - \mathbf{A}_{:,j}|_t$ and $\frac{\partial \mathcal{L}}{\partial \mathbf{B}_{i,:}} \propto \mathbf{B}_{i,:}|_{t-1} - \mathbf{B}_{i,:}|_t$, respectively. Subsequently, we substitute $(\mathbf{BA})_{i,j}$ to Eq. (7) and obtain

$$\begin{aligned}\frac{\partial \mathcal{L}}{\partial (\mathbf{BA})_{i,j}} &\propto [\mathbf{B}_{i,:}\mathbf{A}_{:,j} - (\mathbf{B}_{i,:} - \frac{\partial \mathcal{L}}{\partial \mathbf{B}_{i,:}})(\mathbf{A}_{:,j} - \frac{\partial \mathcal{L}}{\partial \mathbf{A}_{:,j}})], \\ &= \frac{\partial \mathcal{L}}{\partial \mathbf{B}_{i,:}}\mathbf{A}_{:,j} + \mathbf{B}_{i,:}\frac{\partial \mathcal{L}}{\partial \mathbf{A}_{:,j}} - \frac{\partial \mathcal{L}}{\partial \mathbf{B}_{i,:}}\frac{\partial \mathcal{L}}{\partial \mathbf{A}_{:,j}}.\end{aligned} \tag{8}$$

Substitute Eq. (8) to Eq. (6), we can estimate the importance in a gradient-based manner

$$\hat{\mathbf{I}}_{i,j} = ((\frac{\partial \mathcal{L}}{\partial \mathbf{B}_{i,:}}\mathbf{A}_{:,j} + \mathbf{B}_{i,:}\frac{\partial \mathcal{L}}{\partial \mathbf{A}_{:,j}} - \frac{\partial \mathcal{L}}{\partial \mathbf{B}_{i,:}}\frac{\partial \mathcal{L}}{\partial \mathbf{A}_{:,j}})(\mathbf{W}_{i,j} + (\mathbf{BA})_{i,j}))^2. \tag{9}$$

As shown in Figure 2, the LoRA-guided criterion only needs to compute the gradient of $\mathbf{A}$ and $\mathbf{B}$ with the approximation in Eq. (9), which saves memory and computation compared with the gradient of total pre-trained weights $\mathbf{W}_0$. After efficiently estimating the weight importance, we can accumulate the group importance by Eq. (4) for structured pruning.

**Progressive pruning.** To efficiently obtain group importance for structured pruning, we can substitute Eq. (9) into Eq. (4). However, estimating importance and pruning weights with a single batch of

---

**Algorithm 1:** Progressive pruning with LoRA-guided criterion

---

**Require :** Calibration data $\mathcal{D}$; Pre-trained weights $\mathbf{W}_0$; Randomly initialized low-rank matrices
$\quad\quad\quad$ $\mathbf{A}$ and $\mathbf{B}$; Loss function $\mathcal{L}$; Final sparsity level $S$; Fine-tuning iterations $T$.
**Output :** Trained low-rank adaption $\mathbf{A}$ and $\mathbf{B}$; Binary mask $\mathbf{M}$.
$\bar{\mathcal{G}}_g^l \leftarrow 0, \mathbf{M}_g^l \leftarrow 1 \;\forall l, \forall g;$ // Initialization for masks and group
$\quad$ importance
$s \leftarrow 0;$ // Initialize sparsity level
**for** $t \in [1, \ldots, T]$ **do**
$\quad$ Clear gradient;
$\quad$ Forward and backward via Eq. (12);
$\quad$ Update $\mathbf{A}$ and $\mathbf{B}$ via AdamW;
$\quad$ Calculate $\hat{\mathbf{I}}|_t$ via Eq. (9);
$\quad$ Calculate $\hat{\mathcal{G}}|_t$ via Eq. (4);
$\quad$ Calculate $\bar{\mathcal{G}}|_t$ via Eq. (10);
$\quad$ **for** $l \in [1, \ldots, L]$ **do**
$\quad\quad$ $p \leftarrow \text{SortDescending}(\bar{\mathcal{G}})_s;$ // Set threshold
$\quad\quad$ $\mathbf{M}_g^l \leftarrow 0$ where $\bar{\mathcal{G}}_g^l \leq p$, and $g \in [1, \ldots, G]$ // Remove unimportant groups
$\quad$ **end**
$\quad$ Progressively increase $s$ until $||\mathbf{M}||_0 > S$;
**end**

---

data can lead to significant bias and performance loss. To mitigate this, we apply moving averages to evaluate group importance $\mathcal{G}$ and incrementally prune less critical groups. Specifically, the group importance at $t$-th iteration is computed as follows:

$$\bar{\mathcal{G}}|_t = \lambda \bar{\mathcal{G}}|_{t-1} + (1 - \lambda)\hat{\mathcal{G}}|_t. \tag{10}$$

Here, $\bar{\mathcal{G}}|_t$ denotes the group importance score calculated by Eq. (9) and Eq. (4) at the $t$-th iteration, and $\lambda \in [0, 1]$ balances the importance between historical and current values.

To this end, we can efficiently and accurately estimate the importance of each group. We then prune the unimportant groups by setting a binary mask $\mathbf{M} \in \{0, 1\}^{1 \times G}$ for each pruned layer. The binary mask $\mathbf{M}$ is given by

$$\mathbf{M}_g = \begin{cases} 1 & \bar{\mathcal{G}}_g > p \\ 0 & \bar{\mathcal{G}}_g \leq p \end{cases}, \tag{11}$$

where $1 \leq g \leq G$ denotes the $g$-th group in the layer, $p$ represents the threshold of importance. Groups falling below this threshold will be pruned. After setting the mask, the forward process of each pruned layer can be written as

$$\mathbf{z} = (\mathbf{x}\mathbf{W}_0 + \mathbf{x}\mathbf{B}\mathbf{A}) \odot \mathbf{M}, \tag{12}$$

where $\odot$ denotes Hardamard product and can be calculated by broadcast. The complete algorithm of LoRAPrune is given in Algorithm 1.

## 4 EXPERIMENTS

### 4.1 EXPERIMENTAL SETUP

**Models and metrics.** Our method is applied to the LLaMA-1 model family (Touvron et al., 2023), which comprises LLaMA-7B, LLaMA-13B, LLaMA-30B and LLaMA-65B. Following Sun et al. (2023) and Frantar & Alistarh (2023), we evaluate with 2048-token segments for LLaMA-13B, LLaMA-30B and LLaMA-65B. All models are evaluated on the perplexity metric with WikiText (Merity et al., 2016) and PTB (Marcus et al., 1993) dataset. To assess the zero-shot ability of LPMs, we follow LLaMA to perform zero-shot task classification on common sense reasoning datasets: BoolQ (Clark et al., 2019), PIQA (Bisk et al., 2020), HellaSwag (Zellers et al., 2019), WinoGrande (Sakaguchi et al., 2021), ARC-easy (Clark et al., 2018), ARC-challenge (Clark et al., 2018), OpenbookQA (Mihaylov et al., 2018) and MMLU (Hendrycks et al., 2020). We offer the

Table 2: Zero-shot performance of the compressed LLaMA models. We evaluate WikiText2 and PTB on perplexity with 2048-token segments. The average accuracy is calculated among seven classification datasets. **Bold**/Underline denotes the best performance at the same compression rate with/without fine-tuning, respectively. $\star$ denotes the results obtained by our reproduction.

| Pruning Ratio | Method | WikiText2↓ | PTB↓ | BoolQ | PIQA | HellaSwag | WinoGrande | ARC-e | ARC-c | OBQA | Average↑ |
|---|---|---|---|---|---|---|---|---|---|---|---|
| Ratio = 0% | LLaMA-7B (Touvron et al., 2023) | 5.69 | 8.93 | 73.18 | 78.35 | 72.99 | 67.01 | 67.45 | 41.38 | 42.40 | 63.25 |
| Ratio = 20% w/o tune | Random | 20.76 | 38.85 | 61.83 | 71.33 | 56.26 | 54.46 | 57.07 | 32.85 | 35.00 | 52.69 |
| | Magnitude $\star$ | 15.63 | 28.10 | 61.93 | 69.89 | 58.87 | 55.12 | 56.93 | 32.54 | 36.10 | 53.05 |
| | WANDA$\star$ (Sun et al., 2023) | 15.22 | 24.90 | 64.93 | 70.14 | 58.12 | 55.39 | 56.63 | 33.98 | 35.43 | 53.23 |
| | LLM-Pruner (Ma et al., 2023) | 14.36 | 21.82 | 57.06 | 75.68 | 66.80 | 59.83 | 60.94 | 36.52 | 40.00 | 56.69 |
| | LoRAPrune-8bit (Ours) | 14.80 | 22.01 | 57.23 | 74.41 | 65.91 | 59.79 | 61.34 | 34.71 | 39.87 | 56.18 |
| | LoRAPrune (Ours) | 14.74 | 21.80 | 57.98 | 75.11 | 65.81 | 59.90 | 62.14 | 34.59 | 39.98 | 56.50 |
| Ratio = 20% w/ tune | Magnitude $\star$ | 9.06 | 13.80 | 61.89 | 70.81 | 58.34 | 56.87 | 54.87 | 34.02 | 38.40 | 53.59 |
| | WANDA$\star$ (Sun et al., 2023) | 8.64 | 12.66 | 65.75 | 74.70 | 64.52 | 59.35 | 60.65 | 36.26 | 39.40 | 57.23 |
| | LLM-Pruner (Ma et al., 2023) | 8.14 | 12.38 | 64.62 | 77.20 | 68.80 | 63.14 | 64.31 | 36.77 | 39.80 | 59.23 |
| | LoRAPrune-8bit (Ours) | 7.70 | 11.91 | 65.37 | 76.65 | 69.41 | 63.78 | 65.45 | 36.12 | 39.50 | 59.46 |
| | LoRAPrune (Ours) | 7.63 | 11.87 | 65.62 | 79.31 | 70.00 | 62.76 | 65.87 | 37.69 | 39.14 | 60.05 |
| Ratio = 50% w/o tune | Random | 2481.66 | 4107.40 | 46.79 | 53.37 | 27.50 | 50.59 | 28.07 | 27.90 | 30.00 | 37.75 |
| | Magnitude $\star$ | 138.96 | 877.50 | 44.10 | 54.98 | 31.27 | 52.93 | 38.76 | 27.50 | 29.67 | 39.88 |
| | WANDA $\star$ (Sun et al., 2023) | 93.61 | 276.10 | 45.13 | 55.54 | 31.37 | 55.87 | 39.43 | 25.76 | 30.12 | 40.46 |
| | LLM-Pruner (Ma et al., 2023) | 58.83 | 147.11 | 52.32 | 59.63 | 35.64 | 53.20 | 33.50 | 27.22 | 33.40 | 42.13 |
| | LoRAPrune-8bit (Ours) | 68.07 | 163.48 | 50.83 | 56.17 | 34.84 | 53.80 | 33.12 | 27.96 | 31.88 | 41.23 |
| | LoRAPrune (Ours) | 56.30 | 164.96 | 51.78 | 56.90 | 36.76 | 53.80 | 33.82 | 26.93 | 33.10 | 41.87 |
| Ratio = 50% w/ tune | Magnitude $\star$ | 18.36 | 21.88 | 47.40 | 54.36 | 33.49 | 53.10 | 37.88 | 26.60 | 30.12 | 40.42 |
| | WANDA $\star$ (Sun et al., 2023) | 17.38 | 21.34 | 50.90 | 57.38 | 38.12 | 55.98 | 42.68 | 34.20 | 38.78 | 45.43 |
| | LLM-Pruner (Ma et al., 2023) | 16.41 | 20.85 | 60.28 | 69.31 | 47.06 | 53.43 | 45.96 | 29.18 | 35.60 | 48.69 |
| | LoRAPrune-8bit (Ours) | 12.38 | 17.50 | 61.43 | 70.88 | 47.65 | 55.12 | 45.78 | 30.50 | 35.62 | 49.56 |
| | LoRAPrune (Ours) | 11.60 | 17.39 | 61.88 | 71.53 | 47.86 | 55.01 | 45.13 | 31.62 | 34.98 | 49.71 |

Table 3: Runtime results of the structured pruned LPMs.

| Model | Unmerged time (s) ↓ | Merged time (s) ↓ | Perplexity ↓ | Ratio (%) |
|---|---|---|---|---|
| | 0.184(+0.0%) | 0.105(+0.0%) | 5.69 | 0 |
| LLaMA-7B | 0.120(-34.8%) | 0.079(-24.7%) | 7.63 | 20 |
| | 0.089(-51.6%) | 0.053(-49.5%) | 11.60 | 50 |

pruning results on the instruction dataset Alpaca (Taori et al., 2023) in Appendix C. Furthermore, we extend the applicability of the LoRA-guided criterion to unstructured pruning on ViT-B (Dosovitskiy et al., 2020) and BERT-Base (Devlin et al., 2018) due to their relatively small scale and memory requirements. Additional details can be found in Appendix D.

**Implementation details.** We randomly sample 20k sentences from the C4 (Raffel et al., 2020) and 400k sentences from RedPjama (Computer, 2023) dataset, each having a length of 512 tokens. Our training configuration includes a batch size of 128, a learning rate set to 1e-4, and a total of 2 training epochs. As the pre-trained weights remain frozen, there is the option to quantize them into 8-bit values to save memory. All models are optimized by AdamW optimizer (He et al., 2020) with a cosine learning rate decay. These experiments are rigorously conducted on one A100 GPU (80G). When conducting pruning with fine-tuning, we iteratively prune the model until the desired level of sparsity is reached. This process is guided by a cubic sparsity scheduler (Sanh et al., 2020). Conversely, in cases where pruning is performed without fine-tuning, we follow Ma et al. (2023), which randomly samples a batch of data to estimate importance once. Afterward, the model is one-shot pruned, with no weight updates taking place.

**Contenders.** We compare LoRAPrune with the following pruning methods in both fine-tuning and without fine-tuning settings: **1)** Magnitude Pruning: iterative pruning based on the absolute values of model weights. **2)** Random Pruning: iterative pruning with randomly selected weights. **3)** LLM-Pruner (Ma et al., 2023): one-shot pruning using criterion in Eq. (3). **4)** WANDA (Sun et al., 2023): one-shot pruning based on the magnitude of input features and pre-trained weights.

## 4.2 MAIN RESULTS

**Zero-shot performance.** Tables 2 and 5 demonstrate the effectiveness of our proposed method. In situations where LoRAPrune does not undergo fine-tuning to restore model accuracy, its results are comparable to LLM-Pruner which prunes using complete gradients. For example, at a 20% compression rate, LLM-Pruner achieves an average accuracy of 56.69% across seven different inference datasets, while LoRAPrune achieves an average accuracy of 56.50%. However, when fine-tuning is applied to recover model accuracy, our LoRAPrune far surpasses any existing large model pruning methods under structured sparsity. For instance, at a 50% compression rate, LoRAPrune achieves a perplexity of 11.60 on WikiText2, significantly outperforming LLM-Pruner's perplexity

Table 4: Perplexity results on RedPajama corpus.

| Method | WikiText | PTB | Ratio |
|---|---|---|---|
| LLaMA-7B | 5.69 | 8.93 | 0% |
| One-shot Pruning | 6.67 | 10.41 | 20% |
| **Iterative Pruning** | **6.32** | **9.85** | 20% |
| One-shot Pruning | 9.74 | 13.94 | 50% |
| **Iterative Pruning** | **8.31** | **10.83** | 50% |

Table 5: Accuracy on MMLU (5-shot).

| Method | Accuracy (%) | Ratio |
|---|---|---|
| LLaMA-7B | 32.01 | 0% |
| LLM-Pruner | 28.34 | 20% |
| LoRAPrune (Ours) | **29.56** | 20% |
| LLM-Pruner | 26.12 | 50% |
| LoRAPrune (Ours) | **28.03** | 50% |

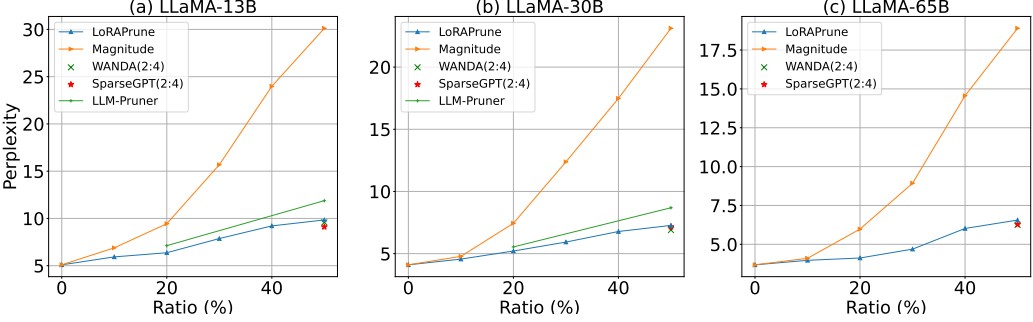

Figure 3: Pruning results on large-scale LPMs (a) LLaMA-13B, (b) LLaMA-30B, (c) LLaMA-65B.

of 16.41. We also replicate the experimental results of WANDA under structured pruning scenarios. Our findings reveal that the pruning outcomes achieved by WANDA fell short in comparison to gradient-based pruning methods such as LLM-Pruner and LoRAPrune. This observation underscores the superior performance and effectiveness of gradient-based pruning approaches in our experiments.

It's worth noting that LoRAPrune's efficient approximation for the gradients of the pre-trained weights allows for 8-bit quantization of those weights, greatly reducing the memory requirements for pruning. Moreover, LoRAPrune demonstrates superior pruning results even when models are quantized to 8 bits. These findings underscore the effectiveness and versatility of LoRAPrune in achieving impressive pruning results across various scenarios and compression rates.

**Acceleration for pruned LPMs.** Models with structured pruning can be directly sped up in general GPU devices. We conducted tests with 2048 tokens, averaging the results over 100 trials. We specifically examined the inference time with and without merging LoRA weights into the pre-trained weights. As shown in Table 3, we observed that when pruning 20% weights, LPM without merging LoRA has an even slower inference speed than LPM with LoRA merged without pruning. In addition, through structured pruning, the model achieves reductions in inference time of 24.7% and 49.5% at compression rates of 20% and 50%, respectively.

**Pruning on large-scale LPMs.** Due to the efficient approximation of the pre-trained weights' gradients, LoRAPrune enables iterative pruning on larger-scale LPMs. To ensure that all experiments can be conducted on one GPU, we quantize the pre-trained weights of LLaMA-30b and LLaMA-65b to 8 bits. The experimental results are shown in Figure 3. We observe that, in comparison to the magnitude-based method, LoRAPrune exhibits significant superiority across various scales. Furthermore, in comparison to the 2:4 sparsity model, LoRAPrune achieves comparable pruning results at a 50% sparsity rate. However, it is worth noting that the 2:4 sparsity model also faces challenges in direct weight merging with LoRA, resulting in additional computational overhead during inference. Besides, accelerating 2:4 sparsity models requires specialized hardware support, such as NVIDIA GPUs based on the Ampere architecture, which significantly increases the deployment constraints for 2:4 sparsity models.

**Scalability of LoRAPrune.** Our LoRAPrune is complementary to large-scale fine-tuning to mitigate the performance drop. We simply replace the tiny post-training calibration set with the large-scale dataset for fine-tuning. We randomly sample 400k data from RedPajama (Computer, 2023) to compare one-shot and iterative pruning strategies, with the experimental results on Wikitext2 shown in Table 4. Under conditions of large datasets, the issue of significant Perplexity reduction is mitigated. It is noteworthy that even with such data volume, iterative pruning still outperforms one-shot pruning markedly.

Table 6: Pruning resource required by different pruning criteria.

| Model | Pruning criteria | Fine-tuning Throughput ↓ | GPU Memory ↓ | Perplexity ↓ |
|---|---|---|---|---|
| LLaMA-7B (Ratio=50%) | Vanilla | 38.87s/iter (+0.0%) | 38.6G (+0.0%) | 11.48 (+0.0%) |
| | Magnitude | 13.08s/iter (-66.3%) | 16.8G (-56.7%) | 17.38 (+52.9%) |
| | LoRA-guided | 14.13s/iter (-63.6%) | 18.3G (-52.6%) | 11.60 (+1.0%) |
| | LoRA-guided (8-bit) | 15.63s/iter (-59.8%) | 13.8G (-64.2%) | 12.38 (+9.0%) |

Figure 4: Similarity between LoRA gradient and vanilla criterion on (a) Attention, (b) MLP layers.

## 4.3 ABLATION STUDY

**Efficiency of LoRA-guided criterion vs. vanilla criterion.** We conduct a comparative analysis of different pruning criteria with respect to their resource requirements and computational efficiency, including GPU memory and throughput. We adopt the vanilla criterion, as outlined in Eq. (3), as our baseline. For each forward pass, we set the batch size to 1, and we accumulate this process iteratively until we reach a total of 128 accumulations. To ensure robustness and reliability, we compute and subsequently average the results obtained over a span of 100 steps. The comparison results can be found in Table 6. Compared to the vanilla criterion, LoRA-guided and LoRA-guided (8bit) criteria demonstrate a significant reduction in GPU memory usage, saving 52.6% and 64.2% of the memory, respectively. Moreover, as the LoRA-guided criterion does not require the computation of original gradients, it achieves a 64.6% increase in throughput compared to the vanilla criterion with comparable performance, greatly enhancing the speed of the pruning process.

**Efficacy of LoRA-guided criterion vs. vanilla criterion.** Since the LoRA-guided criterion in Eq. (9) is an efficient approximation of the vanilla criterion in Eq. (3), we evaluate the effectiveness of the proposed LoRA-guided criterion by comparing mask similarity with the vanilla criterion. We randomly sample 128 data and then perform one-shot pruning with both LoRA gradient and vanilla criterion. Figure 4 illustrates that in the case of low compression rates (Ratio=10%), the masks generated by these two criteria exhibit a high degree of consistency. As the compression rates increase, the mask similarity may decrease. However, it is crucial to emphasize that LoRAPrune follows an iterative pruning approach. In each pruning iteration, it only needs to precisely identify the least important weights (about top-5%), thus ensuring the accurate approximation. Hence, the LoRA-guided criterion can attain pruning results that are on par with those of the vanilla criterion while incurring reduced costs.

## 5 CONCLUSION

In this paper, we have proposed a method to effectively prune and fine-tune LPMs simultaneously, achieving state-of-the-art efficiency-accuracy trade-offs. Specifically, we have proposed a novel LoRA-guided criterion, for evaluating the parameter importance by only computing the LoRA gradients, which greatly reduces the computational resources required for pruning LPMs. Building upon the proposed criterion, we have presented LoRAPrune, a technique that performs efficient joint pruning and fine-tuning without the need for computing gradients of the pre-trained weights. Finally, comprehensive experiments on various LPMs and benchmarks have demonstrated the superiority of LoRAPrune over other pruning methods. In terms of comparison with the vanilla criterion, the LoRA-guided criterion shows its efficiency and effectiveness. In the future, we aim to further enhance the pruning results of LoRAPrune at higher compression rates.

**Limitation.** LoRAPrune requires fine-tuning to restore model performance. This limitation can restrict the application of LoRAPrune in scenarios where fine-tuning is unavailable.

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

# Appendix

## A  WEIGHT DEPENDENCY FOR LLaMA

Here, we use LLaMA architecture as an example to explain the weight dependency. In terms of the Attention module, when we decide to prune a specific head of weights in the Query layer, it is imperative that the corresponding weights with the same index in the Key, Value and Out layers are also pruned. Similarly, for the Feed-Forward Network (FFN) module, when pruning a particular channel of weights in the Up layer, it is essential to prune the weights with matching indices in the Gate and Down layers. This meticulous coordination ensures that pruning maintains the structural integrity and functionality of the model. Following Ma et al. (2023) and Fang et al. (2023), we prune heads for Attention and channels for FFN, respectively. The dependency details are shown in Figure 5.

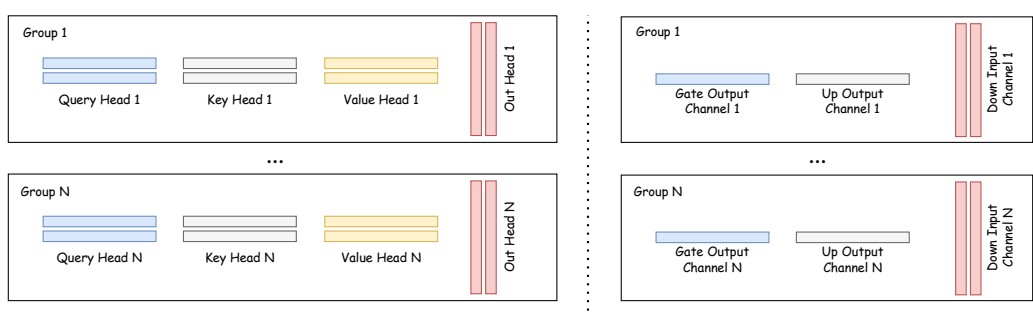

Figure 5: Weight dependency in (a) Attention layer, (b) FFN layer.

## B  MORE ABLATION STUDIES

**Effectiveness of the moving average.** We verify the rationale behind the moving average through the setting of different values for $\lambda$. These experiments were conducted on LLaMA-7b, with experimental configurations consistent with the above implementation details. The experimental results, as shown in Figure 6 (a), reveal that as $\lambda$ increases, the pruning results exhibit a significant reduction in perplexity. This effect is especially pronounced when $\lambda = 0$ where pruning is solely determined by the importance of the current batch, confirming the effectiveness of the moving average.

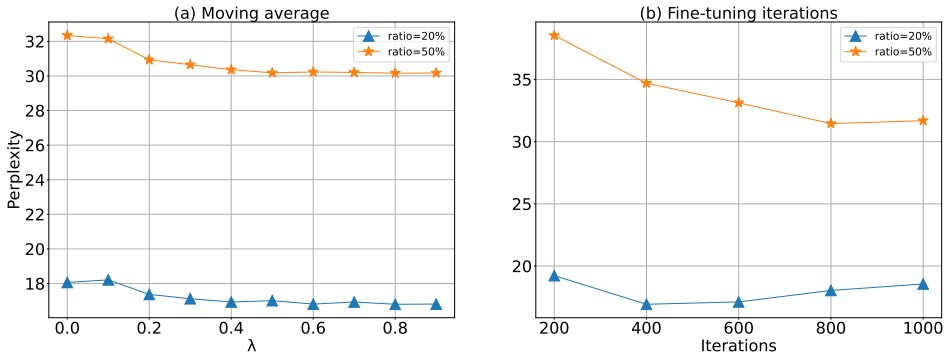

Figure 6: More ablation studies for pruning hyper-parameters: (a) $\lambda$ value in moving average, (b) fine-tuning iterations.

**Impact of iterations.** To assess the impact of the total fine-tuning iterations on pruning results, we conducted experiments on the LLaMA-7b model with different iterations. The results are shown in Figure 6 (b), which indicates that excessive iterations can lead to a decrease in the model's zero-shot performance, potentially due to overfitting on the calibration dataset. Furthermore, we observe that

Table 7: Perplexity results on 128 token segments.

| Method | WikiText2 | Ratio% |
|--------|-----------|--------|
| LLaMA-7B | 12.36 | 0% |
| LLM-Pruner | 17.58 | 20% |
| LoRAPrune | 16.80 | 20% |
| LLM-Pruner | 38.12 | 50% |
| LoRAPrune | 30.12 | 50% |

Table 8: Efficiency comparison between LoRAPrune and LLM-Pruner with CPU off-loading.

| Method | Throughput (s/iter) | GPU Memory (GB) | FLOPs (G) |
|--------|---------------------|-----------------|-----------|
| LLM-Pruner | 38.87 | 38.6 | 20298 |
| LLM-Pruner + CPU offloading | 115.67 | 19.5 | 20298 |
| **LoRAPrune (Ours)** | **14.13** | **18.3** | 12881 |

the model requires more iterations to regain its performance when pruning with high compression (*e.g.*, ratio=50%).

**Impact of calibration data quantity.** We investigate the influence of the calibration data quantity on pruning results by sampling varying quantities of calibration data from the C4 dataset (Raffel et al., 2020). The experimental results indicate that when the calibration data is limited in quantity, the pruning results are unsatisfactory. For instance, with a calibration dataset of only 500 samples, the results of iterative pruning and fine-tuning are not as good as one-shot pruning. As the quantity increases, the pruning results gradually improve. The experiments demonstrate that our default samples (20000 data) are sufficient for fine-tuning and pruning of LPMs.

**Impact of token segments length.** To verify if LoRAPrune maintains its superiority with short token segments, we use 128 token segments to test the perplexity of LLaMA-7B on WikiText2. The results, as shown in Table 7, indicate that LoRAPrune's performance surpasses that of LLM-Pruner even at a 128 token length.

**LoRAPrune vs. LLM-Pruner with gradients off-loading.** The gradient off-loading strategy can partially mitigate LLM-Pruner's memory demands, such as transferring certain gradients to CPU memory. However, the memory access cost and computational overhead are substantial. As illustrated in Table 8, the efficiency gains achieved by our method in LLaMA-7B. In terms of throughput, LoRAPrune is $2.75\times$ faster than LLM-Pruner with offloading and $8.19\times$ faster without offloading. This enables us to use iterative pruning to mitigate the drop arising from structured sparsity.

**Joint vs. separate.** In order to assess the impact of separating the fine-tuning and pruning phases in LoRAPrune, we conduct an experiment that prune LLaMA-7b in one-shot and subsequently employ LoRA to restore its performance. The experimental results presented in Table 10 indicate that joint pruning and fine-tuning yields much better performance than the separate counterpart, especially under the high compression ratio.

**Pruning frequency.** We explore the impact of different pruning frequencies, *i.e.*, how many iterations of fine-tuning before pruning, on the final performance. The experimental results, as shown in Table 11, indicate that our default frequency (frequency=10) obtains the best pruning result. Additionally, we observe that if pruning is too frequent (frequency=1), the model may not have enough iterations to recover through fine-tuning, leading to inaccurate importance estimation. Furthermore, excessive fine-tuning between pruning iterations (frequency=20) leads to overfitting on the calibration data.

Table 9: Pruning results under different calibration data quantity.

| Quantity | Fine-tune | Perplexity with ratio=20% | Perplexity with ratio=50% |
|----------|-----------|---------------------------|---------------------------|
| 128 | ✗ | 20.67 | 121.96 |
| 500 | ✓ | 23.56 | 144.63 |
| 5000 | ✓ | 18.01 | 35.36 |
| 10000 | ✓ | 17.48 | 33.18 |
| 20000 | ✓ | 16.80 | 30.12 |
| 30000 | ✓ | 16.72 | 29.88 |

## C    LORAPRUNE ON INSTRUCTION DATASET

Fine-tuning LPMs on instruction dataset (Taori et al., 2023; Chenghao Fan & Tian, 2023) have demonstrated a remarkable ability to generalize zero-shot to new tasks. Here, we validate the

Table 10: Effect of the joint pruning and fine-tuning. "Average" represents the average performance on seven classification datasets.

| | Method | WikiText2↓ | PTB↓ | Average↑ |
|---|---|---|---|---|
| Ratio=20% | Joint | **16.80** | **28.75** | **60.05** |
| | Separate | 19.87 | 33.69 | 57.18 |
| Ratio=50% | Joint | **30.12** | **38.68** | **49.71** |
| | Separate | 39.60 | 65.79 | 47.56 |

Table 11: Results under different pruning frequencies. "Average" denotes the average performance on seven classification datasets.

| Frequency | WikiText2↓ | PTB↓ | Average↑ |
|---|---|---|---|
| 1 | 31.38 | 40.02 | 49.37 |
| 5 | 30.67 | 39.10 | 49.16 |
| 10 | **30.12** | **38.68** | **49.71** |
| 20 | 31.06 | 38.93 | 49.59 |

Table 12: Generated examples from the Pruned models

| Model | Generated Sentences |
|---|---|
| LLaMA-7b (ratio=0%) | Instruction: Tell me about alpacas. Response: Alpacas are members of the camelid family and are native to the Andes Mountains of South America. They are known for their soft, luxurious fleece, which is used to make clothing, blankets, and other items. Alpacas are herd animals and live in groups of up to 20 individuals. They are generally docile and easy to care for, making them popular as pets and livestock. Alpacas can live up to 20 years in captivity, and their fleece can be sheared up to three times a year. |
| | Instruction: Translate the sentence 'I have no mouth but I must scream' into Spanish. Response: I no tengo boca pero tengo que gritar. |
| | Instruction: Write a Python program that prints the first 10 Fibonacci numbers. Response:
`def fibonacci(n):`
`    if n == 0:`
`        return 1`
`    if n == 1:`
`        return 1`
`    else:`
`        return fibonacci(n - 1) + fibonacci(n - 2)`
`for n in range(1, 10):`
`    print(fibonacci(n))` |
| LLaMA-7b (ratio=20%) | Instruction: Tell me about alpacas. Response: Alpacas are a type of camel-like mammal native to the Americas. They are related to camels and llamas, but are much smaller in size. Alpacas are herbivores and graze on grass and other vegetation. They are known for their gentle and friendly nature, and are often kept as pets or as part of a farm. Alpacas can live up to 10-15 years, and their offspring can live up to 10-15 years as well. |
| | Instruction: Translate the sentence 'I have no mouth but I must scream' into Spanish. Response: Yo no tengo la boca pero tengo que gritar. |
| | Instruction: Write a Python program that prints the first 10 Fibonacci numbers. Response:
`def fibonacci(n):`
`    if n == 0 or n == 1:`
`        return 1`
`    else:`
`        return fibonacci(n - 1) + fibonacci(n - 2)`
`for n in range(1, 10):`
`    print(fibonacci(n))` |

generalization capability of the LoRAPrune by simultaneously fine-tuning and pruning LPMs on the Alpaca (Taori et al., 2023) dataset. The experimental setting follows Alpaca-lora [1]. The experimental instructions encompass common sense, translation, and coding tasks. We show the generative results in Table 12 and observe that even when pruning 20% of the weights, the model continues to excel in performing these tasks.

# D EXTENDING LOAR-GUIDED CRITERION TO UNSTRUCTURED PRUNING

Since ViT-B (Dosovitskiy et al., 2020) and BERT-Base (Devlin et al., 2018) has significantly fewer parameters compared to the LLaMA models, its impact on memory consumption is relatively minor. Therefore, we sought to validate the scalability of LoRA-guided criterion by applying unstructured pruning to ViT-B (Dosovitskiy et al., 2020) and BERT-Base (Devlin et al., 2018). Specifically, similar to the PST (Li et al., 2022b) approach, we begin by computing the LoRA weights **BA** and then generate a binary mask during each forward pass. This mask is applied not only to the **BA** but also simultaneously to the pre-trained weights, introducing unstructured sparsity to both sets of weights.

**Models and metrics.** For image classification tasks, we employ our method on ViT-B (Dosovitskiy et al., 2020) with various pretraining techniques, including supervision on ImageNet-21k (Dosovitskiy

---

[1]https://github.com/tloen/alpaca-lora/

et al., 2020), unsupervision on MAE (He et al., 2022) and MoCOV3 (Grill et al., 2020). We evaluate the pruned ViTs in VTAB-1k (Zhai et al., 2019) which consists of 19 few-shot classification datasets.

For natural language understanding tasks, we employ BERT-base (Devlin et al., 2018) as the pre-trained model. The GLUE benchmark (Wang et al., 2018) is used, which consists of nine natural language understanding (NLU) tasks including natural language inference, text entailment, sentiment analysis, and semantic similarity, among others. The benchmark comprises CoLA, SST-2, MRPC, STS-B, QQP, MNLI, QNLI, RTE.

**Implementation details.** For ViT-B, we set the batch size, learning rate, and weight decay as 64, 1e-3, and 1e-4, respectively. For BERT-Base, we set the batch size to 32 and perform a hyperparameter search over learning rate $\in \{3e\text{-}5, 5e\text{-}5, 1e\text{-}4, 5e\text{-}4\}$ and epoch $\in \{20, 40\}$. All models are optimized by AdamW optimizer (He et al., 2020) with cosine learning rate decay. All experiments are conducted on one NVIDIA RTX 3090 GPU.

**Contenders.** Due to the lack of pruning works conducted under PEFT settings, we replicate several pruning methods: **1)** Magnitude pruning (MaP) (Li et al., 2018) computes the importance of parameters based on their magnitude, making it a data-free pruning method. **2)** Magnitude pruning with LoRA (MaP-LoRA) prunes parameters according to its magnitude and fine-tunes by LoRA. **3)** Movement Pruning (MvP) (Sanh et al., 2020) derives importance from first-order information, making it a data-driven pruning method. **4)** Random Pruning (RaP) (Li et al., 2022a) randomly selects parameters to prune and fine-tunes by LoRA. Both original MaP and MvP are pruned and tuned on the pre-trained parameters. **5)** Parameter-efficient sparsity (PST) (Li et al., 2022b) uses extra low-rank matrices to learn the gradients of pre-trained parameters.

**Image classification.** Firstly, our proposed LoRAPrune outperforms other pruning methods on both FGVC and VTAB-1k datasets, as shown in Table 13. For example, on the 19 tasks of the VTAB-1k dataset, LoRAPrune achieved 72% average Top-1 accuracy that was 4.3% higher than MvP, which prunes using the original parameter gradients. This is because MvP requires fine-tuning of the original parameters during pruning, which can lead to overfitting with limited training data. Moreover, compared to MvP, our LoRAPrune only requires 0.75% of the total parameters to be computed during pruning and fine-tuning, which is much less than MaP and MvP methods. When compared with other PEFT methods such as PST (Li et al., 2022b), MaP-LoRA, and RaP, our LoRAPrune achieves a higher average Top-1 accuracy by 2.9%, 1.7%, and 16.8%, respectively, demonstrating the effectiveness of our proposed LoRA gradient criterion. Secondly, compared to fine-tuning methods without pruning, LoRAPrune produces competitive results. For instance, on the VTAB-1k dataset, LoRAPrune significantly outperforms Linear and Partial-1, and is on par with the VPT.

In addition to the backbones pre-trained with ImageNet-21k, we experiment with two self-supervised methods: MAE (He et al., 2022) and MoCo v3 (Chen et al., 2021). The results are shown in Table 14 and we observe that under the self-supervised pre-trained models, LoRAPrune exhibited remarkably impressive pruning results. For instance, LoRAPrune lags only behind unpruned LoRA-8 using MAE pre-trained weights. LoRAPrune even outperforms LoRA-8 using MoCO v3 pre-trained weights.

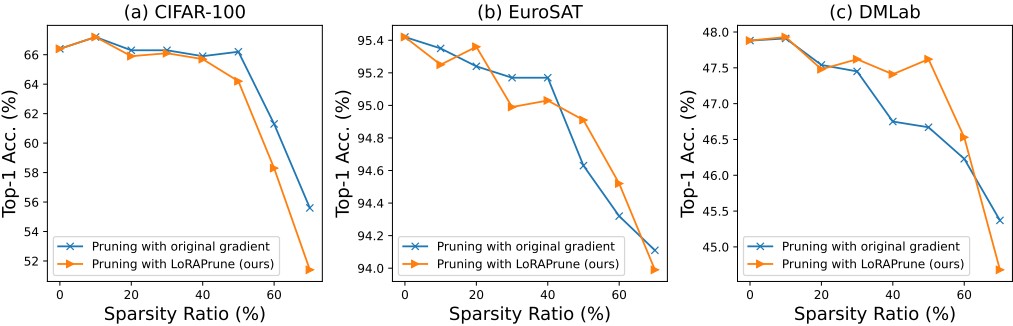

Figure 7: Comparison between LoRAPrune and original gradient-based pruning methods across different sparsity ratios using (a) CIFAR-100, (b) EuroSAT, and (c) DMLab datasets.

Table 13: Comparisons on FGVC and VTAB-1k using ViT-B/16 pre-trained on ImageNet-21k. The sparsity ratio denotes the ratio of pruned parameters, and "Tuned/Total" denotes the fraction of trainable parameters. The best result is in **bold**, and the second-best result is underlined.

| ViT-B/16 (85.8M) | Sparsity ratio | FGVC | | VTAB-1k | | | | |
|---|---|---|---|---|---|---|---|---|
| | | Tuned / Total | Mean Acc. | Tuned / Total | Natural | Specialized | Structured | Mean Acc. |
| Full | 0% | 100.00% | 88.5 | 100.00% | 75.9 | 83.4 | 47.6 | 69.0 |
| **Pruning methods** | | | | | | | | |
| MaP (Li et al., 2018) | 50% | 100.00% | 84.6 | 100.00% | 71.3 | 81.7 | 45.9 | 66.3 |
| MvP (Sanh et al., 2020) | 50% | 100.00% | 84.9 | 100.00% | 72.8 | 80.5 | 49.8 | 67.7 |
| RaP (Li et al., 2022a) | 50% | 0.75% | 73.1 | 0.75% | 53.8 | 70.4 | 41.3 | 55.2 |
| MaP-LoRA | 50% | 0.75% | 83.1 | 0.75% | 75.4 | 81.9 | 53.6 | 70.3 |
| PST (Li et al., 2022b) | 50% | 2.14% | 85.1 | 2.14% | 73.4 | **82.7** | 51.3 | 69.1 |
| LoRAPrune (Ours) | 50% | 0.75% | **85.4** | 0.75% | **76.6** | 82.4 | **57.1** | **72.0** |
| **Unpruning methods** | | | | | | | | |
| Linear | 0% | 0.12% | 79.3 | 0.04% | 68.9 | 77.2 | 26.8 | 57.6 |
| Partial-1 | 0% | 8.38% | 82.6 | 8.30% | 69.4 | 78.5 | 34.2 | 60.7 |
| VPT (Jia et al., 2022) | 0% | 0.75% | 88.4 | 0.75% | 78.5 | 82.4 | 55.0 | 72.0 |
| LoRA-8 (Hu et al., 2022) | 0% | 0.55% | 86.0 | 0.23% | 79.5 | 84.6 | 60.5 | 74.9 |
| LoRA-16 (Hu et al., 2022) | 0% | 0.90% | 84.8 | 0.69% | 79.8 | 84.9 | 60.2 | 75.0 |
| SPT-LoRA (He et al., 2023) | 0% | 0.41% | **89.3** | 0.31% | **81.5** | **85.6** | **60.7** | **75.9** |

Table 14: Comparisons on VTAB-1k using self-supervised ViT-B/16 pre-trained by MAE and MoCo v3. The sparsity ratio denotes the ratio of pruned parameters, and "Tuned/Total" denotes the fraction of trainable parameters. The best result is in **bold**, and the second-best result is underlined.

| ViT-B/16 (85.8M) | Sparsity ratio | VTAB-1k MAE | | | | | VTAB-1k MoCo v3 | | | | |
|---|---|---|---|---|---|---|---|---|---|---|---|
| | | Tuned / Total | Natural | Specialized | Structured | Mean Acc. | Tuned / Total | Natural | Specialized | Structured | Mean Acc. |
| Full | 0% | 100% | 59.3 | 79.7 | 53.8 | 64.3 | 100% | 72.0 | 84.7 | 42.0 | 69.6 |
| Linear | 0% | 0.04% | 18.9 | 52.7 | 23.7 | 32.1 | 0.04% | 67.5 | 81.1 | 30.3 | 59.6 |
| Partial-1 | 0% | 8.30% | 58.4 | 78.3 | 47.6 | 61.5 | 8.30% | 72.3 | **84.6** | 47.9 | 68.3 |
| Bias (Zaken et al., 2022) | 0% | 0.13% | 54.6 | 75.7 | 47.7 | 59.3 | 0.13% | 72.9 | 81.1 | 53.4 | 69.2 |
| LoRA-8 (Hu et al., 2022) | 0% | 0.23% | **66.2** | **82.6** | **62.5** | **70.4** | 0.23% | 69.9 | 83.4 | **59.1** | 70.8 |
| LoRAPrune (Ours) | 50% | 0.75% | 61.3 | 78.8 | 58.4 | 66.2 | 0.75% | **71.5** | 82.5 | 58.6 | **70.9** |

**Natural language understanding.** Table 15 demonstrates the effectiveness of our proposed method. Compared with full fine-tuning methods (MaP and MvP), LoRAPrune achieves comparable or even superior performance to them while only fine-tuning 2.14% parameters. Compared with PST (Li et al., 2022b) that maintains the same number of trainable parameters as LoRAPrune, LoRAPrune achieves an average score improvement of 1.1% on the GLUE dataset when the sparsity ratio is 50%. These results clearly show that our method outperforms existing methods in terms of both model compactness and performance.

**Effect of the LoRA-guided criterion.** We also evaluate the effectiveness of the LoRA gradient criterion in the unstructured setting by comparing it with the vanilla criterion which uses original gradients of pre-trained weights. It is worth noting that to ensure the fairness of the experiments, we only update the LoRA weights for both using the LoRA-guided and vanilla criterion.

We conducted experiments on multiple pruning scenarios with different sparsity ratios using three types of datasets: Natural, Specialized, and Structured. The experimental results, as shown in Figure 7, demonstrate that LoRAPrune achieves comparable or even superior performance to the original gradient-based methods on EuroSAT (Specialized) and DMLab (Structured) datasets. In the case of CIFAR-100 (Natural) dataset, LoRAPrune exhibits competitive performance compared to the original gradient-based methods. These findings validate the effectiveness of the LoRA-guided criterion.

Table 15: Comparisons on GLUE using BERT-Base. The sparsity ratio denotes the ratio of pruned parameters, and "Tuned/Total" denotes the fraction of trainable parameters. The best result is in **bold**, and the second-best result is underlined.

| BERT-Base (110.0M) | Sparsity ratio | GLUE | | | | | | | | | |
|---|---|---|---|---|---|---|---|---|---|---|---|
| | | Tuned / Total | MNLI | QQP | QNLI | SST-2 | CoLA | STS-B | MRPC | RTE | Mean Acc. |
| Full | 0% | 100.00% | 84.7 | 87.8 | 91.5 | 93.0 | 58.6 | 88.7 | 89.5 | 62.9 | 82.0 |
| MaP (Li et al., 2018) | 50% | 100.00% | **83.6** | **87.8** | **91.5** | 91.0 | **60.1** | **89.8** | 90.7 | 67.2 | **82.7** |
| MvP (Sanh et al., 2020) | 50% | 100.00% | 82.3 | 87.3 | 90.8 | 90.8 | 57.7 | 89.4 | **91.1** | 67.2 | 82.1 |
| PST (Li et al., 2022b) | 50% | 2.14% | 81.0 | 85.8 | 89.8 | **91.3** | 57.6 | 84.6 | 90.7 | 67.9 | 81.0 |
| LoRAPrune (Ours) | 50% | 2.14% | 82.4 | 87.2 | 89.6 | 90.9 | 54.1 | 88.7 | 89.8 | **69.3** | 82.2 |
| MaP (Li et al., 2018) | 90% | 100.00% | 78.2 | 83.2 | 84.1 | 85.4 | 27.9 | 82.3 | 80.5 | 50.1 | 71.4 |
| MvP (Sanh et al., 2020) | 90% | 100.00% | **80.1** | 84.4 | **87.2** | 87.2 | 28.6 | **84.3** | 84.1 | 57.6 | 74.2 |
| PST (Li et al., 2022b) | 90% | 2.14% | 79.6 | **86.1** | 86.6 | 89.0 | **38.0** | 81.3 | 83.6 | **63.2** | **75.9** |
| LoRAPrune (Ours) | 90% | 2.14% | 79.4 | 86.0 | 85.3 | **89.1** | 35.6 | 83.3 | **84.4** | 62.8 | 75.7 |

