# OpenReview forum: "LoRAPrune: Pruning Meets Low-Rank Parameter-Efficient Fine-Tuning"
_ICLR.cc/2024/Conference — Submitted to ICLR 2024_

### Official Review · Reviewer_RcVc · 2023-10-23

**Soundness:** 3 good
**Presentation:** 2 fair
**Contribution:** 3 good
**Rating:** 6
**Confidence:** 2

**Summary:**

This paper presents a LoRA-guided structured pruning method for large-scale pre-trained language models. Unlike the previous method that requires gradients w.r.t the entire model parameters, the proposed LoRA-guided criterion only needs to compute the gradients w.r.t up and down low-rank matrices, which can be very lightweight. Based on the method, the authors propose LoRAPrune, which progressively prunes unimportant weights. The experimental results demonstrate that it outperforms the previous LLM-pruning algorithms, given the similar compression rate, while also reducing memory usage.

**Strengths:**

1. LoRA-guided weight importance criterion seems original and interesting
2. Experimental results look very promising, especially when combined with fine-tuning
3. Reduced the resource requirement to do pruning and fine-tuning would constitute a good practical contribution

**Weaknesses:**

1. This paper can benefit from better writing and presentation. A few examples are the following.
1-a.  More details might have been helpful. e.g., what does numbers in Figure 2 mean?
1-b. Abuse of notation, In eq. (6), I_ij, ‘ij’ subscript indicates the index of the matrix I, but in eq 11, I_g, here ‘g’ means the index of the group.
1-c. top-s% has not been formally defined. is it a set?
1-d. In eq 11, I_g \in top-s% -> this notation seems mathematically wrong. I_g probably denotes the importance score.
1-e. In algorithm 1, you calculated I |_t, but it was never used. I think you missed ‘|_t’ in 13th line (inside the double for loop)
1-f. In eq 4, superscripts were used to represent Query, Key, and Value weights. However in algorithm 1, the superscript was used to denote the layer.

2. The authors claim that the proposed method approximates the importance of the weights. Could you present any supporting experimental results on how accurate LoRA-guided importance approximations are? And, following up on this, does better approximation yield better performance eventually?

3. It seems ‘\hat{I}’ has the same dimension as the model weights W? Then, in terms of memory usage (at least if we strictly follow the algorithm 1), is the same as computing the gradients w.r.t. W?

4. It would be helpful for authors to provide ‘theoretical FLOPs’ compared to dL/dW-based importance criterion.

**Questions:**

Questions are embedded in the weakness section.

---

> ### Author Response · Authors · 2023-11-18
> **Response for Reviewer RcVc**
>
> __Q1. This paper can benefit from better writing and presentation.__
>
> Thanks for your suggestion. We have carefully checked and modified the equations and presentation, and reflected these changes in the revised paper.
>
> __Q2. Could you present any supporting experimental results on how accurate LoRA-guided importance approximations are?__
>
> A2. We have already provided the results and discussions in "how accurate LoRA-guided importance approximations are" in the alation study. We set the vanilla gradient-guided criterion in Eq. (3) as our baseline, since our criterion is an efficient approximation of vanilla criterion. As shown in Figure 4, the similarity between these two criteria is very high with Ratio = 10%, which indicates that our LoRA-guided criterion can accurately estimate the importance in an iterative way.
>
> __Q3. Does a better approximation yield better performance eventually?__
>
> A3. It's noteworthy that, compared to the vanilla criterion in Eq. (3), the LoRA-guided criterion is designed for more efficient evaluation of weight importance, not necessarily for enhanced performance. Evidence of this has been presented in Table 10 (Table 5 in the initial submission) and Table 4, which illustrates that while the LoRA-guided criterion maintains performance parity with the vanilla criterion, it significantly reduces the fine-tuning time by 63.6% and GPU memory usage during pruning by 52.6%. This underscores the effectiveness of our method in achieving high-performance levels more efficiently.
>
> __Q4. It seems $\hat{I}$ has the same dimension as the model weights W?__
>
> A4. Yes, $\hat{I}$ has the same dimension as the model weights $W \in R^{d\times k}$. However, we do not need to save $\hat{I}$ for all layers in each iteration. We can calculate $\hat{I}$ group by group. For example, we simultaneously calculate importance of Query, Key and Value in $g$-th Attention layer. Subsequently, we accumulate group importance $\mathcal{\hat{G}}_g \in R^{1\times heads}$  according to Eq. (4) and only  $\mathcal{\hat{G}}_g$  needs to be saved, where $heads << d\times k$.
>
> __Q5. It would be helpful for authors to provide ‘theoretical FLOPs’ compared to dL/dW-based importance criterion.__
>
> A5. We provide the theoretical FLOPs between LoRA-guided criterion and vanilla gradient-guided (dL/dW-based) criterion in the following table. The FLOPs computation rule can be referred to [1]. The table shows LoRA-guided criterion has lower theoretical FLOPs than Vanilla, as it does not compute pretrained weight gradients. We add this comparison in Table 8 in the revised paper.
>
> | Method      | FLOPs (G) |
> |-------------|-----------|
> | LoRA-guided | 12881     |
> | Vanilla     | 20298     |
>
> [1] Scaling laws for neural language models. Arxiv 2020

---

> ### Author Response · Authors · 2023-11-20
> **Happy to provide additional clarification**
>
> We sincerely thank you again for your great efforts in reviewing this paper. We have addressed your major concerns about the presentation and provided theoretical FLOPs compared to the dL/dW-based importance criterion. Please don’t hesitate to let us know if there are still concerns/questions.

---

> > ### Comment · Reviewer_RcVc · 2023-11-22
> > **response and final opinion**
> >
> > Thanks for your thorough response and additional information. I have read your response and other reviews. I think this paper has technical contributions, but I also agree with other reviewers' opinions at the same time. I will keep my score as it was.

---

### Official Review · Reviewer_C5GK · 2023-10-26

**Soundness:** 2 fair
**Presentation:** 2 fair
**Contribution:** 3 good
**Rating:** 5
**Confidence:** 5

**Summary:**

The paper introduces LoRAPrune, a novel framework designed to efficiently compress large pre-trained models (LPMs) for cost-effective inference. LoRAPrune achieves this by introducing a LoRA-guided pruning criterion that utilizes weights and gradients from LoRA, avoiding the memory overhead associated with gradients from pre-trained weights, and a structured iterative pruning procedure.

**Strengths:**

+ The proposed LoRAPrune can achieve practical speedup by introducing the structured sparsity onto the pre-trained large model and the LoRA update.
+ The proposed approach can efficiently guide the pruning process using the LoRA-guided gradient.
+ This paper unifies the parameter-efficient fine-tuning and structured pruning, efficiently saving memory usage.
+ Experiments show that this method works well in practice, and the figures are easy to understand.

**Weaknesses:**

- This paper lacks the motivation why the LoRA can considered as the guidance. Instead, the authors just directly show it can estimate the importance of each parameter. It is encouraged to illustrate why this criterion is created. More explanations and analysis are needed.
- The equations in this paper are unclear and not easy to understand, as there are a lot of abnormal superscripts and subscripts.

**Questions:**

What is the reason why the LoRA-guided creation is better than other pruning criteria? What is the motivation?

---

> ### Author Response · Authors · 2023-11-18
> **Response for Reviewer C5GK**
>
> __Q1. This paper lacks the motivation why the LoRA can considered as the guidance. It is encouraged to illustrate why this criterion is created.__
>
> A1. As explained in the introduction, our primary motivation for employing the LoRA-guided criterion is to reduce computational and memory demands during iterative pruning. As depicted in Figure 1(b), the conventional gradient-guided criterion depends on the gradients of tpre-trained weights, necessitating the computation and storage of gradients with a shape of $d\times k$. In stark contrast, the LoRA-guided criterion requires only the gradients of matrices $A$ and $B$  with shapes of $r\times k$ and $d\times r$, respectively, where $r$ is significantly smaller than $d$ and $k$ (often by a factor of a thousand, as in the case of LLaMA-7B with $d=4096$ and $r=8$). This substantial reduction in size translates to a corresponding reduction in computational load and memory usage. The efficacy of the LoRA-guided criterion, as demonstrated by our experimental results in Figure 4, is its capability to accurately gauge the importance of each weight group, mirroring the accuracy of the traditional gradient-guided approach but with much greater efficiency.
>
> We have incorporated the analysis into the METHOD section in the revised paper.
>
> __Q2. The equations in this paper are unclear and not easy to understand, as there are a lot of abnormal superscripts and subscripts.__
>
> A2. We have explained the superscripts and subscripts in General Response Q3. We have reflected these changes in the revised paper.

---

> ### Author Response · Authors · 2023-11-20
> **Happy to provide additional clarification**
>
> We sincerely thank you again for your great efforts in reviewing this paper. We have addressed your major concerns about motivation and presentation. Please don’t hesitate to let us know if there are still concerns/questions.

---

> ### Comment · Area_Chair_HHm9 · 2023-12-04
> **[Important] Response Required to Authors' Rebuttal**
>
> Dear Reviewer C5GK,
>
> As we progress through the review process for ICLR 2024, I would like to remind you of the importance of the rebuttal phase. The authors have submitted their rebuttals, and it is now imperative for you to engage in this critical aspect of the review process.
>
> Please ensure that you read the authors' responses carefully and provide a thoughtful and constructive follow-up. Your feedback is not only essential for the decision-making process but also invaluable for the authors.
>
> Thank you,
>
> ICLR 2024 Area Chair

---

### Official Review · Reviewer_nG8N · 2023-10-30

**Soundness:** 2 fair
**Presentation:** 3 good
**Contribution:** 2 fair
**Rating:** 5
**Confidence:** 4

**Summary:**

The paper introduces a new framework called LoRAPrune, which aims to efficiently compress LLMs. Existing pruning methods designed for LPMs are not compatible with low-rank adaptation (LoRA), which aims to reduce computational costs. LoRAPrune addresses this by using a LoRA-guided pruning criterion that relies on LoRA weights and gradients rather than the gradients of pre-trained weights, reducing memory overhead. It also introduces a structured iterative pruning procedure to remove redundant channels and heads. Experimental results demonstrate that LoRAPrune outperforms existing approaches, achieving a 50% compression rate while reducing perplexity on datasets like WikiText2 and PTB and significantly reducing memory usage.

**Strengths:**

- The presentation is commendably clear, and all the figures in the paper are of high quality.

- The derivations presented in Section 3.2 are particularly intriguing. The authors employ a LoRA-guided criterion to effectively circumvent the need to store the entire $\frac{\partial \mathcal{L}}{\partial\mathbf{W}}$, resulting in significant memory cost savings compared to existing pruning methods, such as LLM-Pruner.

- The evaluation conducted in this paper is impressively thorough.

**Weaknesses:**

My primary concern regarding the paper pertains to the results presented in Table 2, which clearly indicate a significant degradation in Perplexity (PPL). Such a pronounced reduction in PPL threatens the practical utility of the model. When evaluated with a context window of 2048, a PPL degradation of just 1 already surpasses the performance differential between LLaMA-13B and LLaMA-30B. A mere PPL degradation of approximately 0.2 can account for the disparities between LLaMA and Llama-2. Although the PPL values in the paper might not align with the precise window size I used, but a PPL degradation of 4 unquestionably renders the resulting model inconsequential.

Moreover, I find that the overall methodology is very similar to LLM-Pruner. The only distinction lies in the methodology for computing importance metrics. However, I have reservations about the pivotal significance of the 'starting point' for fine-tuning. Concurrent research, such as Wanda and Sheared-Llama, has demonstrated that fine-tuning a pruned model on a relatively extensive corpus (e.g., RedPajama) diminishes the disparities between LLM-Pruner and more sophisticated, optimization-based pruning criteria (refer to the Sheared-Llama's appendix for more insights). Hence, I am skeptical about the true value of the proposed method, especially when the pruned model is meticulously fine-tuned. Notably, the memory constraints of LLM-Pruner can be effectively mitigated through off-loading, such as utilizing CPU memory to store certain gradients.

Disclaimer: I am not requesting the authors to directly compare their work with Wanda and Sheared-Llama, as these papers have been simultaneously submitted to the same venue. Nonetheless, the insights and findings presented in these two papers may offer valuable context for my assessment of this paper.

**Questions:**

Kindly address my comments under the 'Weaknesses' section. My primary concern regarding this paper is the usability of the resulting models. The PPL degradation from pruning 25% weights is even larger than the performance difference between 13B and 30B models in my opinion. In this context, all speedup and memory-saving metrics appear to lack significance.

---

> ### Author Response · Authors · 2023-11-17
> **Response for Reviewer nG8N**
>
> __Q1. Table 2 shows a significant degradation in Perplexity (PPL).__
>
> A1. We thoroughly address this concern in our General Response Q1. It is critical to highlight that our method significantly outperforms both LLM-Pruner and WANDA at token segments of 128 and 2048.
>
> Notably, our LoRAPrune is complementary to large-scale fine-tuning to mitigate the performance drop, by simply replacing the small post-training calibration data with the large-scale dataset for fine-tuning. Following Sheared-LLaMA [4], we use RedPajama corpus (400k data) to iteratively prune LLaMA-7B. The experimental results can be found in our General Response Q1.
>
> __Q2. Difference with LLM-Pruner.__
>
> A2. We have discussed the difference between LoRAPrune and LLM-Pruner in General Response Q2.
>
> __Q3. I have reservations about the pivotal significance of the 'starting point' for fine-tuning, especially when the pruned model is meticulously fine-tuned.__
>
> A3. We first re-emphasize the practical significance of post-training pruning with limited calibration data:
>
> 1) Acquiring large volumes of fine-tuning data for Large Language Models (LLMs) is often infeasible.
>
> 2) The computational burden of fine-tuning LLMs on extensive datasets is substantial, which can limit their practical deployment.
>
> Given these constraints, our research pivots towards post-training pruning using minimal calibration data. In this setting, the iterative pruning is vital for achieving high-compression rates in models—a strategy that has garnered considerable attention in previous pruning literature [1,2,3]. For instance, as presented in Table 6, the 'Vanilla' model—which can be viewed as an iterative variant of LLM-Pruner—demonstrates a significant reduction in PPL when fine-tuned iteratively, down to 29.96 from 38.12, highlighting the efficacy of iterative pruning.
>
> Furthermore, in response to the reviewer’s query, we further include meticulous fine-tuning scenarios. We evaluate the effectiveness of iterative pruning on large-scale dataset (RedPajama corpus [5]) and compare it with one-shot pruning. The experimental results are shown in the following table. It is noteworthy that even with such data volume (400k), iterative pruning still outperforms one-shot pruning markedly. We include these results in Table 4 in the revised paper.
>
> | Method               | Perplexity | Ratio |
> |----------------------|-------------------------|-------|
> | LLaMA-7B           |         5.69         | 0%    |
> | One-shot Pruning           |        6.67         | 20%    |
> | __Iterative Pruning__          |        6.34          | 20%   |
> | One-shot Pruning           |        9.74          | 50%    |
> | __Iterative Pruning__          |        8.31        | 50%   |
>
>
> [1] Unified and progressive pruning for compressing vision-language transformers. Arxiv 2023
>
> [2] Width & depth pruning for vision transformers. AAAI 2022
>
> [3] Deep Compression of Pre-trained Transformer Models. NeurIPS 2022
>
> [4] Sheared LLaMA: Accelerating Language Model Pre-training via Structured Pruning. Arxiv 2023
>
> [5] Redpajama: An open source recipe to reproduce llama training dataset. 2023
>
> __Q4. The memory constraints of LLM-Pruner can be effectively mitigated through off-loading.__
>
> A4. Gradients off-loading results in reduced throughput which can be found in General Response Q2.

---

> ### Author Response · Authors · 2023-11-20
> **Happy to provide additional clarification**
>
> We sincerely thank you again for your great efforts in reviewing this paper. We have addressed your major concerns about significant PPL degradation and the necessity of iterative pruning. Please don’t hesitate to let us know if there are still concerns/questions.

---

> > ### Comment · Reviewer_nG8N · 2023-11-20
> >
> > Thank you for your prompt and detailed response. After carefully considering your explanation, I have decided to revise my score to 5. However, raising it further to 6 is challenging due to the following reasons:
> >
> > 1. While I acknowledge the perplexity improvements achieved through iterative pruning, it's important to note that this technique is not novel and was introduced in 2015 [source](https://arxiv.org/abs/1506.02626). I still believe that the only distinction between your work and LLM-Pruner is the method of selecting salient weights. A fair comparison between LoRAPrune and LLM-Pruner should be made when iterative pruning is applied to both.
> >
> > 2. Regarding this distinction, the new results presented by the authors highlight the necessity of fine-tuning the complete network after pruning. This raises concerns about the motivation behind LoRA-guided salient weight selection. If LoRA is not employed during fine-tuning, the rationale for using its gradients to guide weight selection becomes unclear.
> >
> > 3. I question the practical significance of 'post-training calibration with limited data.' Given the availability of ample open datasets for large-scale fine-tuning (e.g., RedPajama, RefinedWeb), I believe it is more crucial to produce a usable model than to focus on limited data calibration. Comparing the perplexity achieved by LoRAPrune with existing 3-bit weight quantization methods (often deemed unusable) reveals that the current results of LoRAPrune are still unsatisfactory.
> >
> > 4. The reduction in memory and runtime during the pruning stage seems inconsequential when considering the significantly longer duration of the fine-tuning stage.

---

> > > ### Author Response · Authors · 2023-11-21
> > > **Response for Reviewer nG8N (Part 1)**
> > >
> > > Thanks for your helpful comments. We have addressed your concerns as follows:
> > >
> > > __Q1. Iterative pruning is not novel.__
> > >
> > > A. We did not claim iterative pruning itself as a main contribution of our paper,
> > > which has been widely used in the pruning literature [1-3]. In fact, the main novelty of our approach resides in integrating a new LoRA-based pruning criterion with iterative pruning, where in each iteration, **we use LoRA to concurrently perform parameter selection and fine-tuning**. This new iterative pruning scheme allows for a more efficient way to achieve superior performance compared to LLM-Pruner. To our knowledge, our solution is novel and this novelty has been acknowledged by Reviewers nuRr and RcVc.
> > >
> > > [1] Learning both Weights and Connections for Efficient Neural Networks. NeurIPS 2015
> > >
> > > [2] Packnet: Adding multiple tasks to a single network by iterative pruning. CVPR 2017
> > >
> > > [3] Dropnet: Reducing neural network complexity via iterative pruning. ICML 2020
> > >
> > > __Q2. A fair comparison between LoRAPrune and LLM-Pruner should be made when iterative pruning is applied to both.__
> > >
> > > A. In our initial submission, we have compared LoRAPrune (using LoRA-guided criterion) with LLM-Pruner (using vanilla criterion) under iterative pruning in Table 6. Summarized below for ease of reference, LoRAPrune nearly matches LLM-Pruner's PPL results but with a **52.6% memory reduction** and a **63.6% throughput increase**.
> > >
> > > | Method      | Throughput | GPU Memory| PPL|
> > > |-------------|-----------|-----------|-----------|
> > > | LoRAPrune | 14.13s/iter (-63.6%)     | 18.3G (-52.6%)| 11.60|
> > > | LLM-Pruner + iterative pruning     | 38.87s/iter (0.0%)     | 38.6G (0.0%)|11.48|
> > >
> > > __Q3. Regarding this distinction, the new results presented by the authors highlight the necessity of fine-tuning the complete network after pruning. This raises concerns about the motivation behind LoRA-guided salient weight selection. If LoRA is not employed during fine-tuning, the rationale for using its gradients to guide weight selection becomes unclear.__
> > >
> > > A. We would like to clarify that all our new experiments were conducted with fine-tuning using LoRA, thus not demonstrating the necessity of fine-tuning the complete network. A common practice in concurrent pruning works [1-3], including LLM-Pruner, use LoRA for its memory efficiency when fine-tuning large pruned models, like a 50% reduced LLaMA-65B which still needs around 154 GPU memory for complete fine-tuning. Our motivation for the LoRA-guided criterion has been clearly explained in the introduction.
> > >
> > > [1] LLM-Pruner: On the Structural Pruning of Large Language Models. NeurIPS 2023
> > >
> > > [2] LoRAShear: Efficient Large Language Model Structured Pruning and Knowledge Recovery. Arxiv 2023
> > >
> > > [3] Compresso: Structured Pruning with Collaborative Prompting Learns Compact Large Language Models. Arxiv 2023

---

> ### Author Response · Authors · 2023-11-21
> **Response for Reviewer nG8N (Part 2)**
>
> __Q4. I question the practical significance of 'post-training calibration with limited data.__
>
> A. Fine-tuning the pruned models on a large-scale dataset needs prohibitive computing resources, which can be infeasible for research communities with limited resources. For example, Sheared LLaMA [1] uses RedPajama corpus with 50B tokens and costs about 120 GPU days for fine-tuning. Consequently, much work in the compression community concentrates on post-training scenarios [2-4], utilizing only minimal calibration data for pruning and performance restoration.
>
> Besides, large-scale fine-tuning is not viable in privacy-sensitive or data-limited settings, which are frequently encountered in many real-world scenarios such as in medical, finance, and autopilot domains, where obtaining the data is often challenging due to privacy or confidentiality issues. Therefore, there is a huge demand in industry and academia for pruning LLMs with limited data.
>
> Furthermore, our experiments with datasets of 30k and 400k demonstrate that LoRAPrune's performance can be enhanced by scaling up the fine-tuning dataset. Hence, our approach is not constrained by the size of the dataset. In the future, we plan to extend LoRAPrune to a larger RedPajama corpus to further boost its performance.
>
> [1] Sheared LLaMA: Accelerating Language Model Pre-training via Structured Pruning. Arxiv, 2023
>
> [2] Brecq: Pushing the limit of post-training quantization by block reconstruction. ICLR 2021
>
> [3] A fast post-training pruning framework for transformers. NeurIPS 2022
>
> [4] Bayesian bits: Unifying quantization and pruning. NeurIPS 2020
>
>
> __Q5. The reduction in memory and runtime during the pruning stage seems inconsequential when considering the significantly longer duration of the fine-tuning stage.__
>
> A. Since we use iterative pruning and the moving average technique, the importance of weights must be calculated for each batch during fine-tuning. To clearly delineate the time spent on pruning versus fine-tuning, we provide the GPU hours on the c4 corpus (30k) for LLM-Pruner with iterative pruning and LoRAPrune in the following table, based on LLaMA-7B. Pruning time consists of the time spent on computing the importance, mask adjustment and, if needed, gradients of pre-trained weights. In addition, fine-tuning time covers the forward pass, gradients computation and updates of LoRA weights.
>
> | Method      | Total time (hour) | Pruning time (hour) | Fine-tuning time (hour)|
> |-------------|-----------|-------------|-----------|
> | LoRAPrune |         2   |0.2          | 1.8       |
> | LLM-Pruner + iterative pruning | 5.3 |    3.5|  1.8    |
> | LLM-Pruner + iterative pruning  + off-loading   | 25.8 |24|1.8|
>
> The table indicates that LoRAPrune spends only 10% of the total time on pruning, while LLM-Pruner with iterative pruning spends 66% of the time on pruning.

---

> > ### Comment · Reviewer_nG8N · 2023-11-22
> >
> > Thank you for providing additional insights. While I acknowledge that the paper has made significant progress towards meeting the acceptance criteria, I believe that a substantial revision is still necessary. Specifically, the results and comparisons in Table 2 appear to be largely inconsequential, and I suggest making Table 4 the focal point for presenting the main results. I would appreciate clarification on whether the time measurements provided in the author's response are consistently based on the dataset of 400k sentences from RedPajama. If not, these numbers should also be deemed inconsequential. Furthermore, all comparisons with existing methods should be conducted under RedPajama finetuning.
> >
> > Moreover, the comparisons for iterative pruning are not complete. Traditional iterative pruning methods typically initiate the process from lower sparsity, prune the model, and then finetune the pruned model while gradually increasing sparsity. In this instance, I don't anticipate significant throughput or memory overhead (as mentioned in Table 6), as the pruning process is executed only once at the beginning for each target sparsity level and the finetuning stage is much longer.
> >
> > Besides, the authors have yet to address my inquiry regarding the importance of finetuning versus selecting an optimal starting point for finetuning in LLM pruning. This aspect is crucial and should be a focal point of the revision.
> >
> > Based on the aforementioned observations, I recommend a significant revision of the paper followed by resubmission to the next venue. If the finetuning process requires only 400k x 512 = 0.2 billion tokens with LoRA weights update, I encourage the authors to conduct a head-to-head comparison with ShearedLlama in their future submission (their finetuning cost is around 50 billion tokens). I believe that your method could potentially offer a substantial advantage in finetuning cost, **if there is no degradation in PPL**. Additionally, including more comparisons at the 3B-level models would be valuable, particularly given the success of the recently released StableLM-3B.

---

### Official Review · Reviewer_nuRr · 2023-10-31

**Soundness:** 2 fair
**Presentation:** 2 fair
**Contribution:** 2 fair
**Rating:** 5
**Confidence:** 4

**Summary:**

Large pre-trained models (LPMs) like LLaMA and GLM excel in diverse tasks when fine-tuned. While low-rank adaption (LoRA) can cost-effectively fine-tune LPMs, the vast model scale and computational demands remain challenges. Current pruning techniques for LPMs aren't compatible with LoRA due to issues like their use of unstructured pruning and reliance on the gradients of pre-trained weights. Addressing this, the paper introduces "LoRAPrune," a framework that prunes efficiently using a LoRA-guided criterion and an iterative procedure. This approach avoids computing gradients of the pre-trained weights, ensuring more memory-efficient and accurate models. Tests reveal that LoRAPrune outperforms other methods, reducing memory usage and improving performance.

**Strengths:**

- The authors study the compelling research area of integrating LoRA with pruning methods, meticulously examining the challenges inherent in this combination.

- The proposed framework allows for the concurrent application of structured pruning and LoRA.

- The pruning criteria can be guided by LoRA principles which are novel and useful.

- Comprehensive experimental outcomes using LLaMA models are presented.

**Weaknesses:**

- Upon examining Table 2, several concerns arise regarding the experimental results. Notably, when juxtaposed with the PPL of LLaMA-7B, there's a marked degradation in PPL. Even at a relatively modest 20% pruning rate, the PPL increase is evident. One has to question if there are scenarios where such a pronounced PPL drop would be deemed acceptable.

- Furthermore, the results from WANDA also indicate a significant PPL degradation, which seems to contradict the assertions made in the WANDA paper.

- The 50% pruning rate appears to present substantial challenges. What is the overarching conclusion here? Is the implication that a 50% pruning rate might be overly ambitious? Alternatively, do the authors consider the observed performance drop at this rate to be inconsequential?

- For a more comprehensive understanding, it would be beneficial to have the PPL and other metrics as reported by contemporaneous studies.

**Questions:**

Please see the weaknesses above.

---

> ### Author Response · Authors · 2023-11-17
> **Response for Reviewer nuRr**
>
> __Q1. PPL degradation in Table 2. Such a pronounced PPL drop would be deemed acceptable.__
>
> A1. We thoroughly address this concern in General Response Q1. It is critical to highlight that our method significantly outperforms both LLM-Pruner and WANDA at token lengths of 2048 and 128, as shown in Table 2 and Table 7, respectively. In scenarios where a pronounced PPL drop is observed, it is often possible to mitigate this through the use of additional data to refine the pruning results.
>
> __Q2. The results from WANDA also indicate a significant PPL degradation.__
>
> A2. The WANDA study primarily reported on unstructured and semi-structured pruning results. Our focus, as discussed in our paper's introduction, is on structured pruning due to the hardware efficiency limitations associated with unstructured sparse models. To ensure a fair comparison, we replicated the structured pruning results using WANDA's official codebase with necessary modifications. We make our adapted code available in the supplementary materials to strengthen the reproducibility of our results. Consistent with the aim of fairness, we employed WANDA's metrics to calculate the weight importance, subsequently aggregating the group importance as delineated in Equation (4). The specific modifications are documented in 'wanda/lib/prune.py' lines 171-193 for further examination and reproducibility.
>
> __Q3. The 50% pruning rate appears to present substantial challenges.__
>
> A3.In the post-training structured pruning context, especially when limited calibration data is available, a 50% pruning rate indeed presents substantial challenges. However, we significantly outperform other structured pruning methods. To mitigate the PPL degradation in 50% ratio, we also iteratively prune LLMs on large-scale dataset. The details can be found in General Response Q1.
>
> __Q4. It would be beneficial to have the PPL and other metrics as reported by contemporaneous studies.__
>
> A4. We have duly noted the request for comprehensive metrics. In response, Table 2 has reported PPL and accuracy metrics for zero-shot task classification on the common sense reasoning dataset. To further validate the efficacy of LoRAPrune, we have supplemented these with results from the MMLU (5-shot) evaluations, offering a broader performance perspective consistent with contemporaneous studies. We include these results in Table 5 in the revised paper.
>
> | Method               | MMLU(5-shot) | Ratio |
> |----------------------|--------------|-------|
> | LLaMA-7b             | 36.81        | 0%    |
> | LLM-Pruner           | 28.34        | 20%   |
> | __LoRAPrune (Ours)__ | 29.56        | 20%   |
> | LLM-Pruner           | 26.12        | 50%   |
> | __LoRAPrune (Ours)__ | 28.03        | 50%   |

---

> > ### Comment · Reviewer_nuRr · 2023-11-22
> > **Response**
> >
> > Thank you for your detailed responses and the additional experiments you've conducted.
> >
> > However, I still have critical concerns:
> >
> > 1. Degradation in MMLU and PPL: When structured pruning methods result in a notable decline in MMLU or PPL, they should ideally be compared with a smaller model having similar scores. A 50% pruning that leads to a significant drop in these scores raises questions about its efficacy compared to selecting a smaller model or employing other compression techniques like quantization.
> > 2. Comparative Analysis with LLM-pruner: It would be more insightful to compare with the LLM-pruner to determine the threshold at which a given pruning method starts to significantly impact performance. In the case of LLaMA 7b, both LLM-pruner and LoRAPrune lead to a substantial drop in MMLU. What is the compression ratio where performance degradation is negligible?
> > 3. Methods to Recover Scores or PPL: Are there any additional strategies to recover the MMLU score or PPL after pruning? It's imperative for the authors to propose some basic approaches for enhancing the overall score post-pruning.
> >
> > Given these points, I find it challenging to identify practical applications for the proposed method. Therefore, I  retain my initial evaluation score.

---

> > > ### Author Response · Authors · 2023-11-22
> > >
> > > Thanks for your helpful comments. We have addressed your concerns as follows:
> > >
> > > __Q1. Degradation in MMLU and PPL.__
> > >
> > > A. To improve the performance of the pruned model, we pruned LLaMA-7B by 20% on the LaMini instruction dataset [1]. Owing to time limitations, we continued to assess the model pruned at a 50% ratio (3.3B) using the RedPajama corpus. We also offer performance comparisons with other models that have comparable parameters to ours.
> > >
> > >
> > > | Method               | MMLU(5-shot) | Param |
> > > |----------------------|--------------|-------|
> > > | LLaMA-7B |  36.81 | 7B  |
> > > | __LoRAPrune (Ours)__ |  34.73 | 5.4B  |
> > > | Compresso [2]  | 31.90   | 5.4B   |
> > > | __LoRAPrune (Ours)__ | 28.03  | 3.4B   |
> > > | LLM-Pruner[3] | 26.12  | 3.4B   |
> > > |Open-LLaMA-3B-v1 [4] | 27.0  | 3B   |
> > > |INCITE-Base-3B [5] | 27.0 | 3B |
> > >
> > > The experimental results indicate that after iterative pruning on LaMini dataset, the performance loss due to pruning at a 20% compression rate (5.4B parameters) is significantly reduced. At a 50% compression rate (3.3B parameters), LoRAPrune outperforms other manually designed models [4,5], **demonstrating the substantial value of deriving smaller models through the pruning of larger ones.**
> > >
> > > [1] https://huggingface.co/datasets/MBZUAI/LaMini-instruction
> > >
> > > [2] Compresso: Structured Pruning with Collaborative Prompting Learns Compact Large Language Models. Arxiv 2023
> > >
> > > [3] LLM-Pruner: On the Structural Pruning of Large Language Models. NeurIPS 2023
> > >
> > > [4] Openllama: An open reproduction of llama. 2023
> > >
> > > [5] Redpajama-incite-base-3b-v1. 2023a
> > >
> > > __Q2. Comparative Analysis with LLM-pruner.__
> > >
> > > A. In our previous experiments, pruning 20% of the parameters using our method on a large-scale dataset resulted in an acceptable loss in PPL and MMLU score. We will provide LLM-Pruner's results on a large-scale dataset in the future to analyze the negligible pruning thresholds for LLM-Pruner.
> > >
> > > __Q3. Methods to Recover Scores or PPL.__
> > >
> > > A. By simply increasing the data used for pruning and fine-tuning, the model's performance can be further recovered. For example, by applying LoRAPrune on a larger dataset at a 20% compression rate, we can reduce the Perplexity (PPL) from 7.63 to 6.34 and increase the MMLU score from 29.56 to 34.73. For specific experimental details, refer to Q1 in this response and General Response Q1.

---

> ### Author Response · Authors · 2023-11-20
> **Happy to provide additional clarification**
>
> We sincerely thank you again for your great efforts in reviewing this paper. We have addressed your major concerns about significant PPL degradation. Please don’t hesitate to let us know if there are still concerns/questions.

---

> ### Comment · Area_Chair_HHm9 · 2023-12-04
> **[Important] Response Required to Authors' Rebuttal**
>
> Dear Reviewer nuRr,
>
> As we progress through the review process for ICLR 2024, I would like to remind you of the importance of the rebuttal phase. The authors have submitted their rebuttals, and it is now imperative for you to engage in this critical aspect of the review process.
>
> Please ensure that you read the authors' responses carefully and provide a thoughtful and constructive follow-up. Your feedback is not only essential for the decision-making process but also invaluable for the authors.
>
> Thank you,
>
> ICLR 2024 Area Chair

---

### Official Review · Reviewer_pQXa · 2023-11-06

**Soundness:** 2 fair
**Presentation:** 3 good
**Contribution:** 2 fair
**Rating:** 5
**Confidence:** 4

**Summary:**

This paper proposes a new pruning technique, called LoRAPrune, to perform structural pruning on the target LLM and its LoRA adapters at the same time. Specifically, this paper first proposes a LoRA-guided criterion to indicate the weight importance of LLMs, which works better with LoRA. The proposed LoRAPrune pruning technique is built based on this criterion, which unifies PEFT with pruning. Experiment results show that the proposed method achieves better accuracy compared with existing pruning techniques on LLMs.

**Strengths:**

- The target domain of improving the efficiency of LLMs during inference, especially their compatibility with the SOTA tuning methods (e.g., LoRA adapter).
- The proposed method has the potential to alleviate the memory overhead during pruning, which can potentially enable the proposed pruning technique on a wider range of devices and applications.
- The achieved performance improvement over the baseline methods is promising.

**Weaknesses:**

After reading the paper, I have the following concerns and would like to hear from the authors on their justification. I would like to consider revising my rating based on the authors' feedback.
- To the best of my understanding, the novelty of this paper is limited. Specifically, there are some existing explorations on identifying the dependency during structural pruning to maximally preserve the performance after pruning, such as LLM-Pruner. In this paper, the key difference is that the authors propose to shift the computation of dependency from backbone weight in LLMs to LoRA adapters in LLMs. The author may want to further address the novelty here.
- In Table 1, the authors claim that LLM-Pruner does not support tuning. However, as LLM-Pruner also uses a structural pruning technique, it is also compatible with LoRA adapters, which is indicated in the abstract and experiment sections in LLM-Pruner. The authors may want to further justify their claim.
- In Figure 3, the author missed an important baseline, LLM-Pruner. It would help the authors to better understand the performance of the proposed LoRAPruner by adding the LLM-Pruner baseline in the figure.

**Questions:**

Please refer to the weakness part.

---

> ### Author Response · Authors · 2023-11-17
> **Response for Reviewer pQXa**
>
> __Q1. The novelty of this paper is limited.__
>
> A1. We propose the innovative LoRA-guided pruning criteria, which is informed by low-rank adaptation (LoRA) rather than relying on the gradients of pre-trained weights. This efficient criteria enables us to structurally prune 65B LLMs in a single GPU, improving the applicability of pruning on LLMs. We discuss the differences between LoRAPrune and LLM-Pruner in General Response Q2. Reviewers nuRr and RcVc agree with our novelty.
>
> __Q2. Does LLM-Pruner not support tuning by LoRA?__
>
> A2. We apologize for any confusion caused. To clarify, LLM-Pruner supports tuning with LoRA but does not support iterative pruning efficiently. Specifically, LLM-Pruner uses one-shot pruning and then fine-tunes the pruned models by LoRA. The reasons for LLM-Pruner's inefficiency with iterative pruning are discussed in our General Response Q2. The term "Fine-tune" as referenced in Table 1 pertains to the process of iterative pruning which involves simultaneous pruning and fine-tuning to regain performance. To clarify this process and avoid further misunderstanding, we have updated the terminology in Table 1 from "Fine-tune" to "Iterative Pruning" in our revised submission.
>
> __Q3. The author missed an important baseline, LLM-Pruner, in more large scale models.__
>
> A3. We acknowledge the significance of including LLM-Pruner as a baseline for larger models. Due to the limited timeframe of the rebuttal period, we provide results for the 50% pruning ratio. These results are detailed in the table below.
>
> | Model     |Method | PPL   | Ratio |
> |-----------|-------|-------|----------------------|
> | LLaMA-13B |  LLM-Pruner           |11.88 | 50%   |
> |           |  __LoRAPrune (Ours)__ |9.84  | 50%   |
> | LLaMA-30B | LLM-Pruner           |8.69  | 50%   |
> |           | __LoRAPrune (Ours)__ |7.27  | 50%   |
>
> The preliminary results, as shown in above table, indicate that LoRAPrune outperforms LLM-Pruner in both the LLaMA-13B and LLaMA-30B models at a 50% pruning ratio. This reinforces the scalability and effectiveness of our proposed LoRAPrune. We include these results in Figure 3 in the revised paper.

---

> ### Author Response · Authors · 2023-11-20
> **Happy to provide additional clarification**
>
> We sincerely thank you again for your great efforts in reviewing this paper. We have addressed your major concerns about novelty and additional comparison with LLM-Pruner. Please don’t hesitate to let us know if there are still concerns/questions.

---

> ### Comment · Area_Chair_HHm9 · 2023-12-04
> **[Important] Response Required to Authors' Rebuttal**
>
> Dear Reviewer pQXa,
>
> As we progress through the review process for ICLR 2024, I would like to remind you of the importance of the rebuttal phase. The authors have submitted their rebuttals, and it is now imperative for you to engage in this critical aspect of the review process.
>
> Please ensure that you read the authors' responses carefully and provide a thoughtful and constructive follow-up. Your feedback is not only essential for the decision-making process but also invaluable for the authors.
>
> Thank you,
>
> ICLR 2024 Area Chair

---

### Author Response · Authors · 2023-11-17
**General Response (Part 1)**

We thank all reviewers for their valuable feedback. Overall, our work has been well recognized as "The presentation is commendably clear" (Reviwer nG8N), "original and interesting" (Reviwer RcVc), "Experiments show that this method works well in practice" (Reviewer C5GK). We have summarized and addressed the main concerns as follows:

We have carefully considered and responded to the main concerns:

__Q1. Perplexity Degradation is pronounced compared with the full model in Table 2. (Reviewers nG8N, nuRr)__

A1.
1) Our method falls within the post-training pruning setup, where we need to efficiently prune a well-trained uncompressed model, relying solely on a small amount of calibration data, with small computational costs. This setting aligns with the mainstream setting for concurrent LLM pruning methods [1,2,3]. We highlight that our method significantly outperforms LLM-Pruner [1] and WANDA [2], where the performance drop is due to the limited scale of post-training calibration data.

2) We clarify that our PPL for LLaMA-7b was measured with 128-token segments for fair comparison with LLM-Pruner. Here, we follow WANDA [2] and sparseGPT [3] that uses 2048-token segments to evaluate the perplexity of the pruned models. Our additional results on WikiText2 dataset demonstrate that the perplexity degradation can be significantly reduced under 2048 segment length. To enhance the performance of LoRAPrune, we use the PPL results of 2048-token segments to replace the original results of 128-token segments in Table 2, Table 3 and Table 6.

| Method               | PPL | Ratio |
|----------------------|-------------------------|-------|
| LLaMA-7b  (2048-token)  | 5.69                    | 0%    |
| LoRAPrune (2048-token) | 7.63                    | 20%   |
| LoRAPrune (2048-token) | 11.60                   | 50%   |
| LLaMA-7b  (128-token)  | 12.62                    | 0%    |
| LoRAPrune (128-token) | 16.80                    | 20%   |
| LoRAPrune (128-token) | 30.12                   | 50%   |

3) Our LoRAPrune is complementary to large-scale fine-tuning to mitigate the performance drop. We simply replace the tiny post-training calibration set with the large-scale dataset for fine-tuning. We randomly sampled 400k data from Redpajama [4] to iteratively prune LLaMA-7B, with the experimental results on Wikitext2 shown in the following table.

| Method               | Perplexity | Ratio |
|----------------------|-------------------------|-------|
| LLaMA-7B           |         5.69         | 0%    |
| LoRAPrune          |        6.34          | 20%   |
| LoRAPrune          |        8.31        | 50%   |

Under conditions of large datasets, the issue of significant perplexity reduction is mitigated. We include these results in Table 4 in the revised paper.

---

> ### Author Response · Authors · 2023-11-17
> **General Response (Part 2)**
>
> __Q2. Difference from LLM-Pruner. (Reviewers pQXa, nG8N)__
>
> A2. We are the first method that unifies efficient fine-tuning with structured pruning. Our method has fundamental differences with LLM-Pruner [1]:
>
> 1) Our work is the first to integrate LoRA with iterative structured pruning, achieving both parameter-efficient tuning and direct hardware acceleration during inference. In particular, our new LoRA-guided pruning criteria uniquely leverages only LoRA's weights and gradients, making the pruning process memory-efficient and fast. Therefore, LoRAPrune enables scalability to LLaMA-65B on a single GPU. This is in sharp contrast to LLM-Pruner, which uses full gradients for pruning and thus hard to generalize to large-scale models under limited computational and storage resources. The experimental results in Table 2 and Table 8 demonstrate that LoRAPrune outperforms LLM-Pruner by a perplexity reduction of 4.81 on WikiText2 and 3.46 on PTB datasets, while concurrently reducing memory usage by 52.6%.
>
> 2) Although LLM-Pruner’s memory demands can be partially mitigated by off-loading strategies, such as transferring certain gradients to CPU memory, the memory access cost and computational overhead are substantial. The following table clearly illustrates the efficiency gains achieved by our method in LLaMA-7B. In terms of throughput, LoRAPrune is 2.75$\times$ faster than LLM-Pruner with offloading and 8.19$\times$ faster without offloading. This enables us to use iterative pruning to mitigate the drop arising from structured sparsity. We include these results in Table 8 in the revised paper.
>
> | Method                      | Throughput   | GPU Memory |
> |-----------------------------|--------------|------------|
> | __LoRAPrune (Ours)__        | __14.13s/iter__  | __18.3G__      |
> | LLM-Pruner                  | 38.87s/iter  | 38.6G      |
> | LLM-Pruner + CPU offloading | 115.67s/iter | 19.5G      |
>
> Overall, LoRAPrune represents a fundamental advancement over LLM-Pruner, both in technical methodology and in the broader implications of its application.
>
>
> [1] LLM-Pruner: On the Structural Pruning of Large Language Models. NeurIPS 2023
>
> [2] A Simple and Effective Pruning Approach for Large Language Models. Arxiv 2023
>
> [3] SparseGPT: Massive Language Models Can be Accurately Pruned in One-Shot. ICML 2023
>
> [4] Redpajama: An open source recipe to reproduce llama training dataset. 2023
>
>
> __Q3. This paper can benefit from better writing and presentation.__
>
> A3. We list the main modifications as follows:
>
> We polish the notations following the suggestions from Reviewer RcVc. For instance, to avoid confusion with the use of subscripts and superscripts, we revised Equation (4) and now use $\mathcal{G}$ to represent what was originally $\mathbf{I}_g$. Additionally, we outline the notations employed in the equations at the beginning of Section 3.1.

---

> > ### Author Response · Authors · 2023-11-22
> > **Happy to provide additional clarification**
> >
> > We are reaching out to check whether the reviewers have any additional questions based on our previous response.

---

### Meta-Review · Area_Chair_HHm9 · 2023-12-09

**Metareview:**

The paper under review proposes an approach integrating LoRA with pruning methods, focusing on enhancing the efficiency of Large Language Models (LLMs) during inference. The average review score is 5.2, situating the paper at the boundary. While the concept and experimental outcomes are promising, the paper is challenged by concerns about its novelty, the clarity of its presentation, and the practical implications of its findings.

**Justification For Why Not Higher Score:**

Based on the reviews and the average score of 5.2, the recommendation leans toward rejection. Despite the paper's approach in integrating LoRA with pruning methods and promising experimental results, significant concerns persist regarding its novelty, clarity, and practical utility. The issues with the presentation, ambiguity in the methodology, and degradation in performance metrics such as perplexity are concerning. While the authors have made efforts to address some of these issues, the remaining concerns, combined with the borderline score, suggest that the paper may not yet meet the conference's threshold for acceptance.

**Justification For Why Not Lower Score:**

N/A

---

### Decision · Program_Chairs · 2024-01-16

Reject